# Structural Attention Enhanced Continual Meta-Learning for Graph Edge Labeling Based Few-Shot Remote Sensing Scene Classification

**Feimo Li [1]** , **Shuaibo Li [2]** , **Xinxin Fan [3]** , **Xiong Li [4]** and **Hongxing Chang [1,\*]**

1. Institute of Automation Chinese Academy of Sciences, 95 Zhongguancun East Road, Haidian District, Beijing 100190, China; lifeimo2012@ia.ac.cn
2. School of Information, Central University of Finance and Economics, Shunsha Road, Shahe Higher Education Park, Changping District, Beijing 102206, China; 2020212352@email.cufe.edu.cn
3. College of Software, Beihang University, 37 Xueyuan Road, Haidian District, Beijing 100190, China; zf1921334@buaa.edu.cn
4. School of Mechanical Electronic and Information Engineering, China University of Mining and Technology, Ding No.11 Xueyuan Road, Haidian District, Beijing 100083, China; SQT2000401010@student.cumtb.edu.cn
\* Correspondence: hongxing.chang@ia.ac.cn

**Abstract:** Scene classification is one of the fundamental techniques shared by many basic remote sensing tasks with a wide range of applications. As the demands of catering with situations under high variance in the data urgent conditions are rising, a research topic called few-shot scene classification is receiving more interest with a focus on building classification model from few training samples. Currently, methods using the meta-learning principle or graphical models are achieving state-of-art performances. However, there are still significant gaps in between the few-shot methods and the traditionally trained ones, as there are implicit data isolations in standard meta-learning procedure and less-flexibility in the static graph neural network modeling technique, which largely limit the data-to-knowledge transition efficiency. To address these issues, this paper proposed an novel few-shot scene classification algorithm based on a different meta-learning principle called continual meta-learning, which enhances the inter-task correlation by fusing more historical prior knowledge from a sequence of tasks within sections of meta-training or meta-testing periods. Moreover, as to increase the discriminative power between classes, a graph transformer is introduced to produce the structural attention, which can optimize the distribution of sample features in the embedded space and promotes the overall classification capability of the model. The advantages of our proposed algorithm are verified by comparing with nine state-of-art meta-learning based on few-shot scene classification on three popular datasets, where a minimum of a 9% increase in accuracy can be observed. Furthermore, the efficiency of the newly added modular modifications have also be verified by comparing to the continual meta-learning baseline.

**Keywords:** remote sensing scene classification; few shot learning; continual meta-learning; graph transformer

## 1. Introduction

Scene classification is one of the techniques fundamental in many remote sensing tasks with a wide range of applications, such as ecological and hydrological monitoring [1], urban planning [2], forest mapping and conservation [3], resource exploration [4], and agricultural assessments [5]. In recent years, with the development of satellite and aerial imaging technologies, the accumulation of free high quality remote sensing images grows to a huge quantity, and a number of famous domestic and foreign remote sensing research institutes including Wuhan University, Northwestern University of Technology, Beijing Normal University, INRIA, and DLR have published their own public scene classification datasets. Such prosperity in data sharing greatly encourages the development of deep

learning-based methods, and the highest performance has constantly been renewed with the fast publication of novel algorithms [6]. Lately, the performances of algorithms based on traditional whole dataset range training with sufficient sample quantities have almost reached 100%, which is competitive even to human analyzers [7–9].

In the meantime, such advantageous performances come at high supervision costs, for every modification on the model requires a large amount of manual work in data collection and costly model retraining. Such characteristics result in several significant limitations on applications [10]. The first limitation takes place in transferability, for remote sensing scenery images have high intra-class variances caused by differences in regions, seasons, weathers, etc., models for recognizing the same land type also need to be retrained as to prevent large accuracy losses. The second limitation happens in the scalability, where any extension on the recognizable class set of the model requires thorough retraining over all the previous training samples. To make matters worse, if there are not enough samples for the extended class, heavy data collection and annotation labors are required. Many efforts have been made to alleviate these two issues under the traditional training framework, common methods include using the correct model parameter transfer methods to reduce the feature shift in cross domain model application [11], or to augment a dataset with artificial samples generated by generative adversarial network techniques [10].

Due to these problems, a new research topic for model training under data urgent conditions called Few-Shot Learning (FSL) has come into the view. FSL is designed for applicable scenarios where samples are naturally insufficient or very hard to collect, and most of the researches published under this topic adopted the concept of priori knowledge maximization, which happens both in datasets processing and model modifications. So far, the main categories of the state-of-art FSL solutions include: Meta-learning based [12,13], embedding learning-based [12–14], and generative modeling-based [15–18]. These three categories of methods focus on optimizations of the learning mechanism, feature embedding space, and training data samples specifically, and have achieved good applications results in areas that include classification [12,19], regression [20], detection [14,17], segmentation [21,22], etc.

The idea of using few-shot learning algorithms for the remote sensing scene classification problem dates back to 2018 [23–26], where researchers found severe data insufficiency problems in dealing with hyperspectral and SAR images. After that, researches using the similar principles spread quickly to other types of remote sensing images. Among the transfers of algorithms, the dataset extension is a straightforward migration, such as extending the quantity of samples using generative methods [27]. While for some other methods, a very different approach to construct the classifiers has been chosen, as most of have dropped the obsessiveness of making models as robust and accurate as those trained from large datasets, and pursued a tradeoff between their performances and usage scopes by extending the sample sets or building more flexible models. Based on such ideas, many ingenious methods have been devised. For instance, there are methods focused on optimizing the embedded feature space such as feature interpolation [28] and knowledge distillation [29,30] for better sample discrimination. Some methods utilized the heatedly studied attention mechanism such as attention fusion [31,32] and attention metric calculation [33] to improve discriminative power.

Among these FSL-based remote sensing scene classification methods, one category based on meta-learning has received much more attention in recent researches [34,35]. The core idea of meta-learning is learning the model adaptation patterns over specific data distributions, which is helpful for fast model convergence on a small dataset, and is fitted with the fundamental needs of few-shot learning. Categorized by the representative format of prior knowledge, meta-learning methods can be roughly divided into three categories: The meta-representation-based methods [36–38], the meta-optimization-based methods [39–41], and the meta-objective-based methods [42–44]. More recently, newly proposed researches also exploited the combination with metric learning [12,45,46], reinforcement learning, [47–49] and graph models [50–52], which has made a considerable

number of technical breakthroughs in the field. As a result of this, an increasing number of few-shot remote sensing scene classification methods choose to use the meta-learning concepts. For instance, in [53,54], the efficiency of meta-learning methods have been explored on satellite and aerial images, where a novel balance loss and a new feature embedding cosine metric have been proposed specifically to improve the effectiveness of iterative model optimization. Article [55] fused the emerging frequency and imagery content into the process of embedding space optimization, and proposed a new loss objective based on a combination of contrast loss and cross-entropy loss, which greatly improved the convergence speed. Article [56] focused on the trade off between training dataset complexity and size of the model, then proposed a novel parameter trimming and fusion mechanism through parameter transferring, which effectively combine the meta-learning technique with an unsupervised domain adaptation training mode, and improved the training efficiency with fewer model parameters.

To sum up, meta-learning-based methods have shown their capability in fast model convergence on small datasets, but the performance gap between these methods and those trained with sufficient samples via the traditional way still exists. One of the major reasons is that the task-wise dynamic training and inferencing mechanism not only promotes the flexibility of the model, but also increases the isolation between samples in the sequence of tasks, which limit the further improvement in the accuracy of the model. To address this, a novel training mechanism called continual meta-learning [57] is adopted in this article to exploit the informative implicit correlations and alleviate the catastrophic forgetting phenomenon, which facilitates the transfer of historically-encoded node features for the recognition of newly input samples. Additionally, we further utilize structural attention to improve the node feature encoding via the graph transformer [58], which is further combined with an edge labeling Bayesian graph network [52] to further increase the categorical discriminative power.

## 2. Related Works

### 2.1. Continual Meta-Learning

The concept of continual meta-learning derives from standard meta-learning, and is built upon the assumption that the performance of the model will continually increase alongside accumulative meta-learning through tasks. Thus, the model can maintain a well-performing status on newly input samples without extra retraining. Such a mechanism is different from a standard meta-learning mechanism where the dynamic updating range is limited within a single meta-task, continual meta-learning utilizes historical knowledge from the whole learning process, which is especially beneficial under data urgent conditions. However, with the existence of a catastrophic forgetting phenomenon, model performance will not always grow monotonically or stay at a reasonable level, but suffer from significant regressions [59]. To address such a problem, many research efforts have been made. The representative methods include using an attention mechanism in continual learning [60], optimizing the feature-embedded space [36,61], or the formation process of it [62], tweaking the training mechanism [63].

### 2.2. Graph Neural Networks

The graph neural network (GNN) models composed of nodes and edges are natural for non-structural data modeling, since they can map data feature space into non-Euclidean metrics [64]. Thus they have been widely used in applications, including advertising analysis [65], social calculation [66], chemical molecular analysis [67], etc. Recently, GNNs have been frequently used to cater to challenging Few-shot Classification (FSC) problems. Typical examples include transforming the classification problem into edge labeling [57], or modeling complex hierarchical correlations [68]. Moreover, GNNs have also been widely used with meta-learning concepts, which effectively improve the deep model performance on small datasets. For instance, modeling inter-class relationships [51,52], improving feature embedding [51], or using it with continual meta-learning principles.

*2.3. Self-Attention and Graph Transformer*

Attention has always been considered an important discrimination feature in pattern recognition. Currently, researches using a transformer-based attention calculation technique have made significant progress in performances and grow to prosperity. The transformer modules were firstly used widely on natural language processing, equipped with a multi-head attention function, showing expertise in extracting a useful feature from sequentially-ordered inputs. Recent studies have managed to transfer them to vision tasks including classification [69,70], detection [71], segmentation [72], etc. successfully, and indicate a limitless potential in the future. Graph transformers are attention calculation models derived from the transformer models that focus on extracting the significant feature from non-structural data. Such characteristics facilitates the representative graph structure recognition, which is very valuable for applications involving complex scene recognition. Typical applications include paragraph chapters comprehension [58,73], objective relationship recognition [74,75], 2D and 3D spatial structure perception [76], time and space evolution process analysis [77], etc. Technically, the usage of transformer-based graph attention computation improves encoded graph features [58], optimization processes [74], updating mechanisms [75], etc.

## 3. Preliminary

Before diving into the details of the proposed algorithm, we will make a brief introduction to the core concepts of few-shot learning and continual meta-learning, as well as the mathematical definitions and symbols being used in these fields.

**Few-shot classification** is essentially a classification problem itself, where a classifier $f : x \to y$ is to be trained on a given training set $D_{train}$, with $x$ being the input image, and the predicted categorical label $y$ belongs to a set of classes predefined, where $y \in C = \{1, 2, ..., c_n\}_{n=N}$. And under the normal training and testing setting, samples of the training and testing datasets are disjoint, used independently, but selected from the set of classes $C$. That is, $\{D_{train}, D_{test}\} = \{(x_i, y_i)\}_{i=1}^{N}$, $D_{train} \cap D_{test} = \varnothing$, as is shown in Figure 1a. The classifier is optimized and converged on the training set with a large and sufficient quantity of samples, which are the typical characteristics of deep learning methods. However, in few-shot classification problems, samples for training are limited or scarce, with a categorical sample quantity $\left|D_{train}^C\right|$ being very small for each class $C$. Most few-shot learning problems follow a $N$-way $K$-shot specification, trained and tested in units called tasks, where $K$ support samples and $K$ query samples from $N$ classes are selected for training and testing. The $K$ small ranges from 1 to 10 usually, and the trained classifier can only be adjusted on the $K$ samples from the $N$, before making accurate predictions on new test samples. This might seem impossible, but by modeling patterns in similarly distributed samples, or making more abstractive samples correlation modeling, these have helped the latest few-shot classification methods to achieve considerable accuracies, including the applications in remote sensing scenarios.

**Meta-learning** is one of the most developed and widely applied Few-shot Learning (FSL) solutions, its core concept is to learn the learning strategy on a set of tasks from a specific distributions, which makes the model capable of converging efficiently on only a small number of samples from unprecedented environments. In meta-learning, training and testing are organized in units of tasks $\{T_i\}_N$, each having its own training and testing samples called the support and query dataset, noted as $S$ and $T$, and $T = S \cup Q$. Under such dataset organization, for a $N$-way $K$-shot task, there is a support set $S = \{(x_i, y_i)\}_{i=1}^{N \times K} = \bigcup_{n=1}^{N} \{(x_j, y_j), y_j = n\}_{j=1}^{K}$ with a size of $N \times K$, and the query set has $N \times K$ or other quantities of samples from the same collection of $N$ classes. The training and testing processes of the meta-learning are also termed as meta-training or meta-testing, where classes used for training $D_{meta-train}$ and testing $D_{meta-test}$ are disjoint $C_{meta-train} \cap C_{meta-test} = \varnothing$, and models have to deal with unseen classes with the assis-

tance of very scarce supervision. To define such a classification prediction in the form of Bayesian inference, there is:

$$p\left(y_i^Q \middle| x_i^Q, S; \Theta\right) = p\left(y_i^Q \middle| x_i^Q, \psi_S; \Theta\right) p(\psi_S|S; \Theta). \tag{1}$$

In Equation (1), $\Theta$ represents the parameters denoting the priori knowledge learned from training tasks $T \sim p(T)$ with similar distributions as the testing ones, which usually kept static during the testing. $\psi$ represents the task-dependent parameters correlated with the support subset $S$. Evidently, the transferability of a meta-learning model depends on two factors: Firstly, learning a set of more transferable priori parameters and secondly, finding the appropriate task-dependent parameter on the small support sample set.

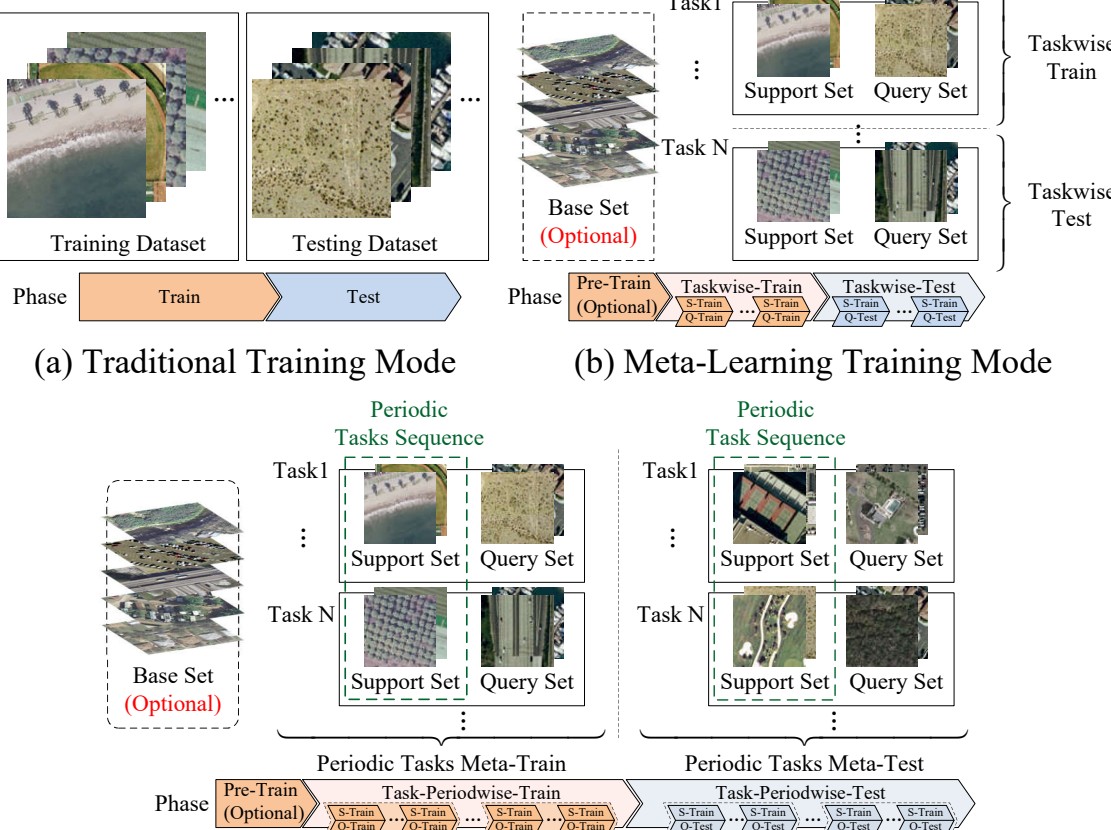

(a) Traditional Training Mode

(b) Meta-Learning Training Mode

(c) Continual-Meta-Learning Training Mode

**Figure 1.** Illustrative comparisons of different training modes. (**a**) Traditional training mode, monolithic train and test periods. (**b**) The typical meta-learning training and testing in units of tasks. (**c**) The typical continual meta-learning training and testing procedure adopted in this paper.

Such a definition of meta-learning greatly encourages the flexibilities of training and testing, but also imports implicit manmade isolation between support sample sets fed sequentially to the model, which has virtually impacted the further improvement in meta-learning algorithm performances. Due to this, a branch of meta-learning research, named Continual Meta-learning (CML), have returned to researching interests, which can make continuous model updates while processing the input tasks sequence $\{T_u|u = 1, ..., U\}$, and eventually affects the task-dependent parameter $\psi_j$. Such a process is like a constant reutilization of the latest historical knowledge, noted as $h_i \sim p(h_i|S_{1:i-1}; \Theta)$, which gives the following equation:

$$p\left(y_i^Q \middle| x_i^Q, S; \Theta\right) = p\left(y_i^Q \middle| x_i^Q, \psi_S, h_i; \Theta\right) p(h_i|S_{1:i-1}; \Theta) p(\psi_S|S; \Theta). \tag{2}$$

In Equation (2), with the continual optimization of the hidden historical parameters $h_i$, the prediction accuracy query sample $x_i^Q$ is improved by the support sets $\{S_{1:i-1}\}$ in all previous processed tasks $\{T_{1:i-1}\}$. Based on such a learning concept, quite a few continual meta-learning algorithms explored the efficiencies of time series data modeling techniques including the Markov chain, LSTM for performance enhancement. Under such efforts, sharing between the learned task-dependent parameters is promoted, and significant algorithmic performance improvements have been seen in applications including scene classification, image segmentation, and object detection.

Finally, the performance advantages of continual learning methods does not come solely from the implicitly increased support samples, but depends more on the correlations being extracted across tasks, which have been used to enhance optimizing the generalization ability of the model on correlated yet unseen data samples. In fact, as being exemplified in our experiments, the incremental degree in overall accuracies surpasses the incremental ratio in sample quantities.

## 4. Overall Framework

As shown in Figure 2, the proposed algorithm architecture consists of three components: The encoding module, the continual learning module, and the Bayesian graph edge labeling module. In the proposed algorithm design, the support and query samples in tasks will be firstly encoded by the ResNet as node features, along with the structural attention features from the graph transformer in the encoding module. Then the node features will be fed to the gated recurrent unit (GRU) based continual meta-learning module for distributional optimization. After that, the distributionally stabilized node features and the structural attention are fed to the Bayesian graph labeling module for node correlation estimation via a Bayesian style edge weight estimation based on Gaussian distribution approximation. The functional design and characteristics of the three modules are described below.

The **encoding module** encodes the primitive features, including node features and structural attention features, for subsequent processing. In it, node features from task-wise support and query samples are primitively encoded by a Convolutional Neural Network (CNN) for multi-scale texture pattern modeling. Then the structural attention is calculated upon the distance adjacency matrix from the CNN features through a graph transformer unit, which helps to rectify the non-optimal categorical measures between different classes. Furthermore, the calculated structural attention is multiplied with the CNN node features for enhancement.

The **continual learning module** tightly connects the graph transformer from the encoding module and the subsequent Bayesian graph labeling module, being more like a macro structure adopting feature embedding improvement. Its core component is a GRU-based sequential data iterative optimization unit, using the long-short term memory-based historical feature fusion mechanism to recall and emphasize the intrinsic local correlations between categorical node features in the sequentially encoded node features. This module is also the central design to alleviate the catastrophic forgetting problem under task-dependent meta-learning framework.

The **Bayesian graph edge labeling module** weighs the node feature correlations and produce the final classification results. In it, an explicit fully connected adjacency graph upon the support and query samples is established. Then a Gaussian distributional approximation of the actual support and query nodes correlations via a Bayesian style inference from the combinatorial usage of the adjacency graph edge weights and the structural attention from the graph transformer, as the modeled estimates of the mean and deviation values of the resulting connection strengths. Finally, the rectified edge weights are transformed into the correlations between query and support samples, and turned into the estimated class labels of the query samples.

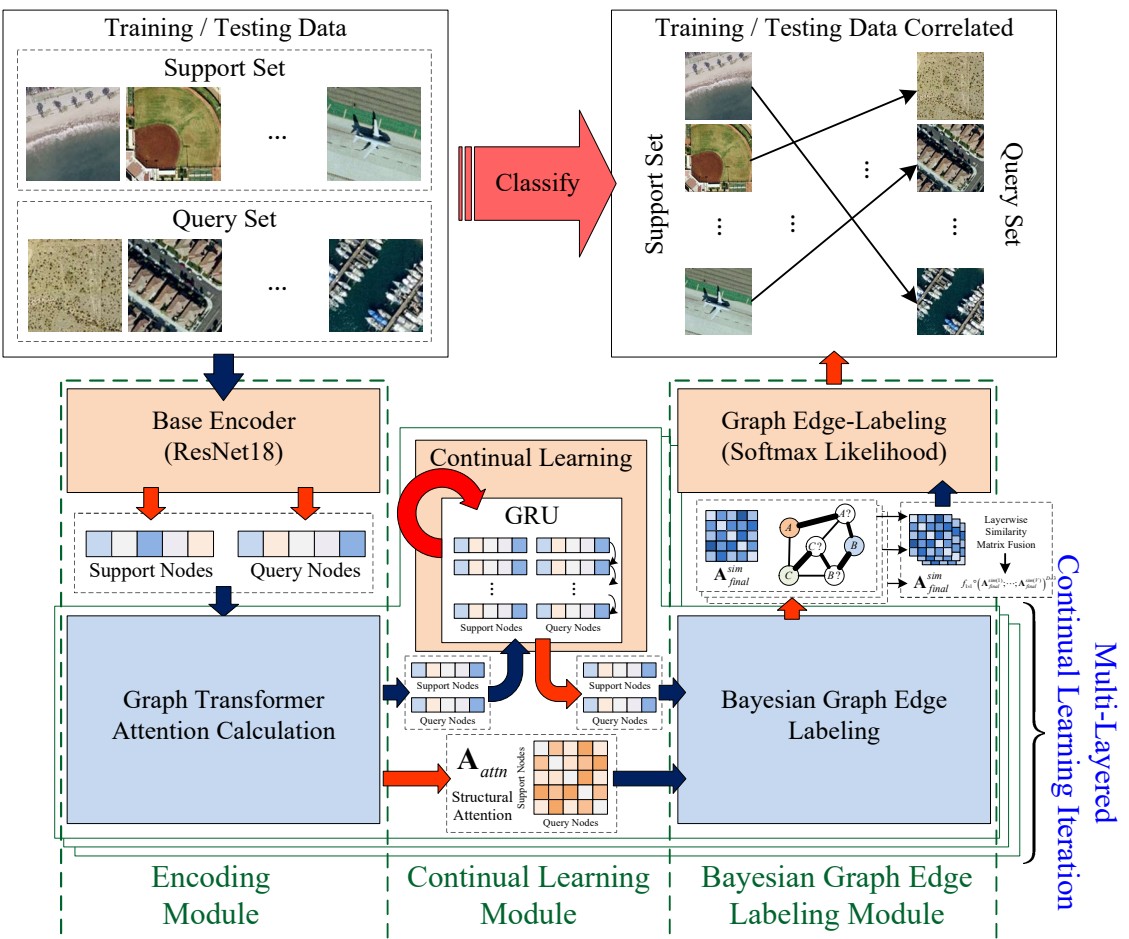

**Figure 2. Overview**: The proposed continual meta-learning few-shot scene classification algorithm is composed of three main components: The encoding module, the GRU-based continual meta-learning module, and the Bayesian graph neural network-based graph edge labeling module.

## 5. Algorithm Details

### 5.1. Encoding Module

Shown in Figure 3, the encoding module consists of two procedures: Node feature-encoding procedure, and the structural attention-encoding procedure. According to symbol definitions in Section 3, the support and query samples in a single task can be denoted as $T = (S, Q)$, and the encoding module extracts the primitive features and structural properties from them.

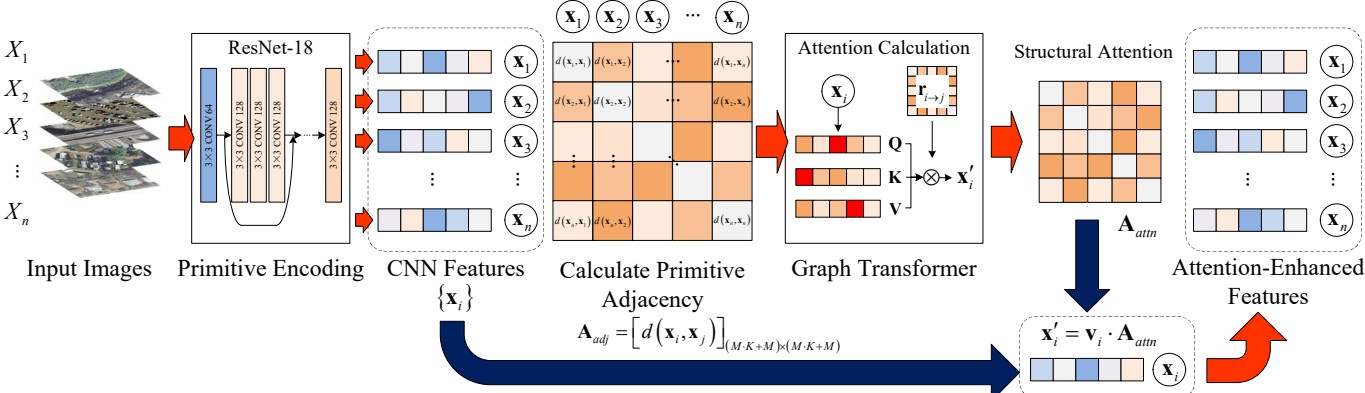

**Figure 3.** Illustration of the processing procedure of the encoding module.

During the node feature encoding procedure, the $N \times K + M$ support and query samples are firstly fed to a CNN to get the encoded features $X_T = X_S \cup X_Q$, where $X_S = \{\mathbf{x}_i | x_i \in S\}$ and $X_Q = \{X_j | \mathbf{x}_j \in Q\}$. Emperically, ResNet-18 is chosen as the CNN backbone, which is adequately complex for preserving the feature differential details and revealing the categorical structure.

Then, in the structural attention encoding procedure, an adjacency matrix $\mathbf{A}_{adj}$ is calculated straight-forwardly at first by getting differences $d()$ between encoded features $X_T$ in Equation (3). In practice, the numerical difference can be norm-1 or norm-2.

$$\mathbf{A}_{adj} = \begin{bmatrix} d(\mathbf{x}_1, \mathbf{x}_1) & \cdots & d(\mathbf{x}_1, \mathbf{x}_s) & \cdots & d(\mathbf{x}_1, \mathbf{x}_{N+M}) \\ \vdots & \ddots & & & \vdots \\ d(\mathbf{x}_t, \mathbf{x}_1) & \cdots & d(\mathbf{x}_t, \mathbf{x}_s) & \cdots & d(\mathbf{x}_t, \mathbf{x}_{N+M}) \\ \vdots & & & \ddots & \vdots \\ d(\mathbf{x}_{N+M}, \mathbf{x}_1) & \cdots & d(\mathbf{x}_{N+M}, \mathbf{x}_s) & \cdots & d(\mathbf{x}_{N+M}, \mathbf{x}_{N+M}) \end{bmatrix}, (\mathbf{x}_t, \mathbf{x}_s) \in X_T \times X_T \tag{3}$$

As mentioned in [58], adjacency matrix $\mathbf{A}_{adj}$ derived from encoded feature value differences $d(\mathbf{x}_t, \mathbf{x}_s)$ might have some problems representing correlations where a categorical structure is complex due to its discriminative power. Thus, a graph structure attention calculation module called a graph transformer is applied here, to enhance the encoding of correlations. In it, the attention matrix $\mathbf{A}_{attn}$ are deducted from the linear transformation of node features $\mathbf{x}_i$, where the attention query $\mathbf{q}_i$, key $\mathbf{k}_i$, and value features $\mathbf{v}_i$ are firstly calculated in Equation (4):

$$\begin{aligned} \mathbf{q}_i &= \mathbf{x}_i \cdot \mathbf{W}_q + \mathbf{b}_q \\ \mathbf{k}_i &= \mathbf{x}_i \cdot \mathbf{W}_k + \mathbf{b}_k \\ \mathbf{v}_i &= \mathbf{x}_i \cdot \mathbf{W}_v + \mathbf{b}_v. \end{aligned} \tag{4}$$

Then, depending on the positional relationship of $\mathbf{q}_i$ and $\mathbf{k}_i$ in the attentional multiplication, a linear transformation operation is applied for constructing the directional correlation adjacency matrix:

$$\begin{aligned} \mathbf{R}^{\rightarrow} &= \mathbf{A} \cdot \mathbf{W}^{\rightarrow} + \mathbf{b}^{\rightarrow} = [\mathbf{r}_{i \rightarrow j}]_{i,j} = [\mathbf{r}_{1 \rightarrow j}, \mathbf{r}_{2 \rightarrow j}, ..., \mathbf{r}_{M+N \rightarrow j}]_j \\ \mathbf{R}^{\leftarrow} &= \mathbf{A} \cdot \mathbf{W}^{\leftarrow} + \mathbf{b}^{\leftarrow} = [\mathbf{r}_{i \leftarrow j}]_{i,j} = [\mathbf{r}_{1 \leftarrow j}, \mathbf{r}_{2 \leftarrow j}, ..., \mathbf{r}_{M+N \leftarrow j}]_j. \end{aligned} \tag{5}$$

The $\mathbf{R}^{\rightarrow} = [\mathbf{r}_{i \rightarrow j}]_{i,j}$ and $\mathbf{R}^{\leftarrow} = [\mathbf{r}_{i \leftarrow j}]_{i,j}$ represents the two kinds of directional correlations in between the support set $S$ and query set $Q$. By this, directional correlations $\mathbf{R}^{\rightarrow}$ and $\mathbf{R}^{\leftarrow}$ are further fused with the query features $\mathbf{q}_i$ by linear operations, which produces the structural attention matrix $\mathbf{A}_{attn}$ as in below:

$$\mathbf{A}_{attn} = \begin{bmatrix} \mathbf{q}_i + \mathbf{r}_{i \rightarrow 1} \\ \mathbf{q}_i + \mathbf{r}_{i \rightarrow 2} \\ \vdots \\ \mathbf{q}_i + \mathbf{r}_{i \rightarrow M+N} \end{bmatrix} \cdot \begin{bmatrix} \mathbf{q}_i + \mathbf{r}_{i \leftarrow 1} & \mathbf{q}_i + \mathbf{r}_{i \leftarrow 2} & \cdots & \mathbf{q}_i + \mathbf{r}_{i \leftarrow M+N} \end{bmatrix}. \tag{6}$$

With the structural attention matrix $\mathbf{A}_{attn}$, and the self-attention weighted vector $\mathbf{v}_i$, the structural attention strengthened node feature $\mathbf{x}'_i$ can be calculated by following:

$$\mathbf{x}'_i = \mathbf{v}_i \cdot \mathbf{A}_{attn}. \tag{7}$$

Lastly, the gotten attention weighted node feature $\mathbf{x}'_i$ will be adaptively combined with the CNN-encoded node features by means of skip connection, which helps the network to increase its flexibility and robustness.

*5.2. Continual Meta-Learning by Online GRU-Based Feature Optimization*

As introduced in the preliminaries in Section 3, continual learning contributes to the model performance not only by the implicit increments in sample quantity, but mainly by the informative sharing between task-wise encoded features, which can be represented by a hidden historical parameter iteratively updated in the process. Following the Bayesian structure, the hidden parameter correlated with the support and query samples in task sequence $\{T_u\}_{u=1}^U$ can be noted as $h_i \sim p(h_i|S_{1:i-1};\Theta)$.

The Gated Recurrent Unit (GRU) is commonly used as a simplified LSTM module, which is computationally more efficient. In the GRU module, the attention strengthened node feature $\mathbf{x}'_i$ is serially encoded, as being shown in Figure 4:

$$
\begin{aligned}
r &= \sigma\big(W_{jr}\mathbf{x}'_i + b_{jr} + W_{hr}h_{i-1} + b_{hr}\big) \\
z &= \sigma\big(W_{jz}\mathbf{x}'_i + b_{jz} + W_{hz}h_{i-1} + b_{hz}\big) \\
n &= \tanh\big(W_{jn}\mathbf{x}'_i + b_{jn} + r \odot (W_{hn}h_{i-1} + b_{hn})\big) \\
h_i &= (1-z) \odot n + z \odot h_{i-1}.
\end{aligned}
\tag{8}
$$

In Equation (8), $z$ can be taken as a controlling parameter of $h_{p-1}$ and $n$, where $h_{p-1}$ is the historical hidden parameter, and n is the converted value from input node feature $\mathbf{x}'_i$. If $z$ grows larger, then more historical information will be used from the hidden variable $h$, otherwise the current value will be adopted more.

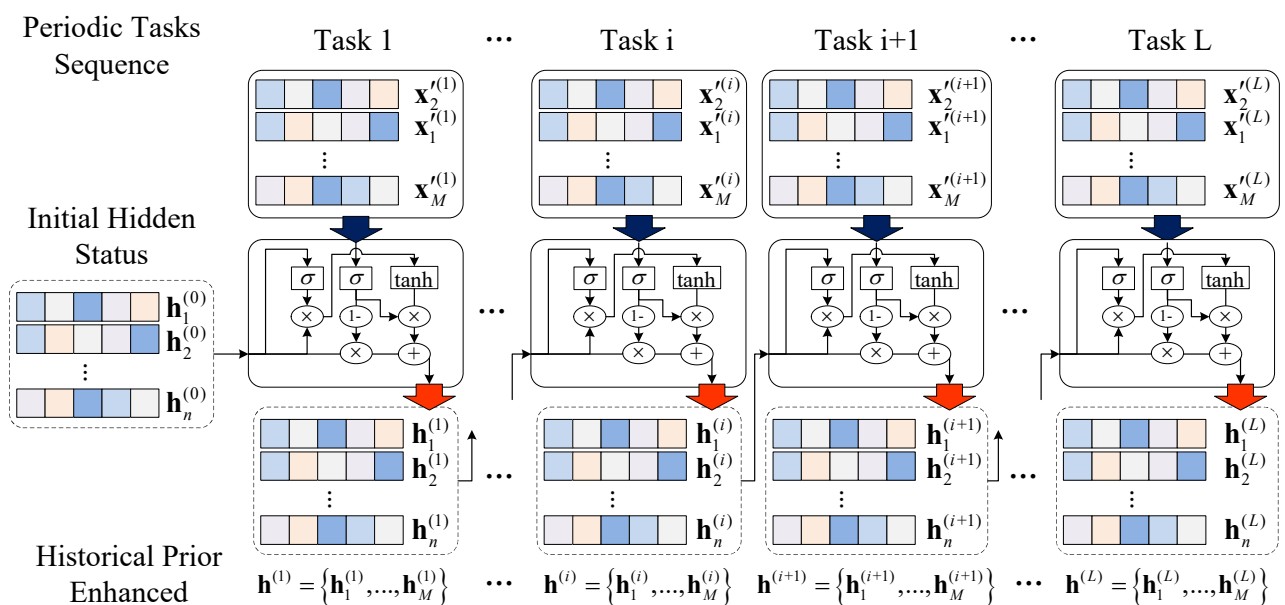

**Figure 4.** Illustration of the GRU-based sequential encoding of features in tasks in the continual meta-learning module.

In the proposed algorithm, such GRU-based node feature polishing is performed through a task sequence sampled from the meta-training data subset, the longer this sequence is, the stronger long-short term memory historical information referencing effect it has. Moreover, the input features $\mathbf{x}'_i$ can be concatenated from two or more tasks to suite the conditions using a larger training batch size, and the node features being packed and fed in a single iteration is called a cell $c_i$. For instance, for training settings with batch size $S_{batch} = 32$ and GRU iteration length $L_{GRU} = 16$, then the input feature to the GRU unit is packed from $C = S_{batch}/L_{GRU} = 2$ tasks, and $c_i = \bigcup_{k=1}^C \{\mathbf{x}'_j | \mathbf{x}' \in T_{i \times C + k}\}$.

Then the sequentially cell-wise-encoded hidden state features $h_i$, are rearranged into task-wise feature collections $\bigcup_{u=1}^U \{\mathbf{h}_i | \mathbf{h}_i = f_{GRU}(\mathbf{x}'_i), \mathbf{x}'_i \in T_u\}$, which are fed to the succeeding Bayesian graph edge Labeling model.

*5.3. Bayesian Graph Edge Labeling for Classification*

In the Bayesian graph edge labeling module, distributionally converged node features are fused with the structural attention features from a graph transformer to produce the final estimates on the correlations between query and support samples for class prediction. This feature fusion takes the form of a Bayesian style inference upon the primitive adjacency matrix from the node feature distances and the structural attention matrix.

Final sample correlation inference takes place via two branches that later merges. The first branch produces the primitive adjacency matrix $\mathbf{A}'_{adj}$, which is calculated in a similar way as the adjacency matrix $\mathbf{A}_{adj}$ in Equation (3) for getting structural attention, as shown in the upper left part of Figure 5. In $\mathbf{A}'_{adj}$, the fundamental node features are the hidden state features $\mathbf{h}_i$ outputted by the GRU-based continual meta-learning module, as in below:

$$
\mathbf{A}'_{adj} = \begin{bmatrix} d(\mathbf{h}_1, \mathbf{h}_1) & \cdots & d(\mathbf{h}_1, \mathbf{h}_{N+M}) \\ \vdots & \ddots & \vdots \\ d(\mathbf{h}_{N+M}, \mathbf{h}_1) & \cdots & d(\mathbf{h}_{N+M}, \mathbf{h}_{N+M}) \end{bmatrix}.
\tag{9}
$$

After that, the feature difference $\mathbf{A}'_{adj}$ is fed to a Fully Convolutional Neural network (FCN) $f^{sim}$ with $1 \times 1$ filter kernels to produce the mean correlation matrix $\mathbf{A}^{sim} = f^{sim} \circ \mathbf{A}'_{adj}$, as the final output of the first branch.

For the second branch in parallel, as has been shown in the lower left part of Figure 5, a set of calibration parameters $\mu_{ij}^{\mathbf{W}}, \delta_{ij}^{\mathbf{W}}, \mu_{ij}^{\mathbf{b}}, \mu_{ij}^{\mathbf{b}}$ are estimated from the $\mathbf{A}_{attn}$ outputted by a graph transformer by a set of FCNs, $f_* = \{f_{M_{\mathbf{W}}}, f_{\Delta_{\mathbf{W}}}, f_{M_{\mathbf{b}}}, f_{\Delta_{\mathbf{b}}}\}$.

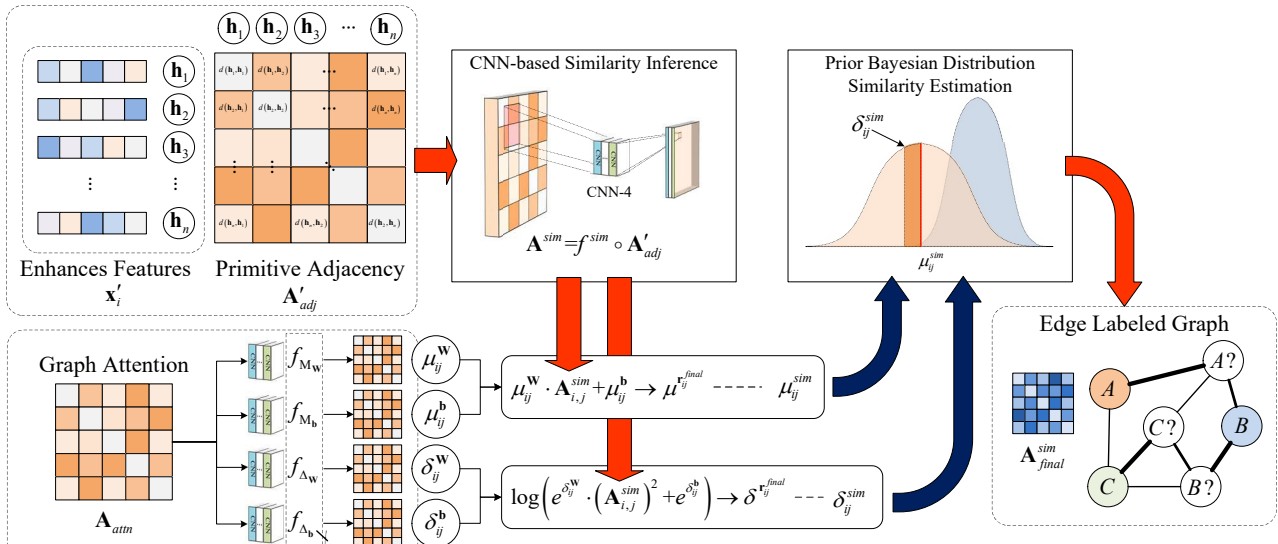

**Figure 5.** Illustration of the calculation workflow in the Bayesian graph labeling module.

The second branch is the core of the Bayesian style correlations estimation concept, where the superscripts of the calibration parameters $\mathbf{W}_t$ and $\mathbf{b}_t$, are supposed to follow a collection of Gaussian distributions as in the equations below:

$$
\mathbf{W}_t \sim N\left(M_{\mathbf{W}}, \Delta_{\mathbf{W}}^2\right), \mathbf{b}_t \sim N\left(M_{\mathbf{b}}, \Delta_{\mathbf{b}}^2\right).
\tag{10}
$$

These calibration weights and biases are calculated by the FCNs in $f_*$ with a $1 \times 1$ filter kernels calculated with the following equations:

$$
\begin{aligned}
\mu_{ij}^{\mathbf{W}} &= f_{M_{\mathbf{W}}} \circ \mathbf{r}_{ij}^{enhc}, \quad \delta_{ij}^{\mathbf{W}} = f_{\Delta_{\mathbf{W}}} \circ \mathbf{r}_{ij}^{enhc} \\
\mu_{ij}^{\mathbf{b}} &= f_{M_{\mathbf{b}}} \circ \mathbf{r}_{ij}^{enhc}, \quad \mu_{ij}^{\mathbf{b}} = f_{\Delta_{\mathbf{b}}} \circ \mathbf{r}_{ij}^{enhc}.
\end{aligned}
\tag{11}
$$

In Equation (11), there are $\mathbf{r}_{ij}^{enhc}$ from $\mathbf{A}_{attn}$, and $M_{\mathbf{W}}=\left\{\mu_{ij}^{\mathbf{W}}\right\}$, $\Delta_{\mathbf{W}}=\left\{\delta_{ij}^{\mathbf{W}}\right\}$, $M_{\mathbf{b}}=\left\{\mu_{ij}^{\mathbf{b}}\right\}$, $\Delta_{\mathbf{b}}=\left\{\delta_{ij}^{\mathbf{b}}\right\}$, as the mean and deviation of the Gaussian-style mapping parameters $\mathbf{W}_t$, $\mathbf{b}_t$.

The attention-enhanced support and query sample correlation $\mathbf{r}_{ij}^{adj}$ from $\mathbf{A}'_{adj}$ will be mapped into the sample similarity matrix $\mathbf{A}_{final}^{sim}$, with the correlation $\mathbf{r}_{ij}^{final}$ from $r_{ij}^{adj}$ weights $\mathbf{W}_t$ and biases $\mathbf{b}_t$.

Finally, for every correlation $\mathbf{A}_{final}^{sim}$ of the resulting similarity matrix $\mathbf{A}_{final}^{sim}$, it is generated via the parameters $\mathbf{W}_t$, $\mathbf{b}_t$, with the equations:

$$
\begin{aligned}
\mathbf{r}_{ij}^{final} &\sim N\left(\mu^{\mathbf{r}_{ij}^{final}}, \delta^{\mathbf{r}_{ij}^{final}}\right) \\
\mu^{\mathbf{r}_{ij}^{final}} &= \mu_{ij}^{sim} = \mu_{ij}^{\mathbf{W}} \cdot \mathbf{A}_{i,j}^{sim} + \mu_{ij}^{\mathbf{b}} \\
\delta^{\mathbf{r}_{ij}^{final}} &= \delta_{ij}^{sim} = \log\left(e^{\delta_{ij}^{\mathbf{W}}} \cdot \left(\mathbf{A}_{i,j}^{sim}\right)^2 + e^{\delta_{ij}^{\mathbf{b}}}\right).
\end{aligned}
\tag{12}
$$

Furthermore, in order to enhance the estimated inter-sampler correlations, annotated as multi-layered continual learning iteration in the lower right side of Figure 2, the correlations calculated by the graph transformer module, GRU module, and the Bayesian graph edge labeling module can be calculated in a multi-layered form. During this multi-layered correlation inference, several (denoted as $V$ here) similarity maps $\left\{\mathbf{A}_{final}^{sim}{}^{(v)}\right\}_{v=1}^{V}$ are generated through $V$ independently created chains of these modules. The final similarity matrix is produced via a FCN with $1 \times 1$ sized kernels, as in Equation (13). In it, $\left(\mathbf{A}_{final}^{sim(1)}; \cdots; \mathbf{A}_{final}^{sim(V)}\right)^{D=3}$ means stacking the similarity matrices at 3rd dimension:

$$
\mathbf{A}_{final}^{sim} = f_{1\times 1} \circ \left(\mathbf{A}_{final}^{sim(1)}; \cdots; \mathbf{A}_{final}^{sim(V)}\right)^{D=3}, \mathbf{A}_{final}^{sim(i)} \in \left\{\mathbf{A}_{final}^{sim(v)}\right\}_{v=1}^{V}.
\tag{13}
$$

Then the sample-wise correlation $\mathbf{r}_{ij}^{final}$ in $\mathbf{A}_{final}^{sim}$ between the support and query set is picked out to be rectified by the logit function $\sigma(\cdot)$ and the query-to-support mask matrices $\mathbf{M}_q$ and $\mathbf{M}_s$, where $\mathbf{M}_q$ will preserve the query sample slots and $\mathbf{M}_s$ preserves the support samples slots, as in the following equation:

$$
p\left(\tilde{y}^{(k)} \mid \tilde{x}, h^{(k)}, \psi_t\right) = \mathbf{M}_q \odot \mathbf{A}_{final}^{sim} \odot \mathbf{M}_s.
\tag{14}
$$

The overall training process of proposed model is illustrated in Algorithm 1. During training, the accuracy of the posterior distribution of query samples inferred from query-to-support correlations, will be updated when comparing with the true query sample labels to construct the graph edge accuracy loss $\mathcal{L}_E$, which will be combined with the basic loss $\mathcal{L}_B$ derived from the other components of the model.

$$
\mathcal{L}_E = -\sum_{k=1}^{K}\sum_{i=1}^{|\mathcal{Q}|} \tilde{y}_i \log\left(p\left(\tilde{y}_i^{(k)}\right)\right) + (1-\tilde{y}_i)\log\left(1 - p\left(\tilde{y}_i^{(k)}\right)\right).
\tag{15}
$$

$$
\Theta^* = \arg\min_{\Theta} \mathcal{L}_E + \gamma \mathcal{L}_B.
\tag{16}
$$

In the above equations, $\Theta^*=\left\{\theta_{ResNet}, \theta_{GraphTrans}, \theta_{GRU}, \theta_h, \theta_E\right\}$ represents the parameters being used across the algorithmic framework, where $\theta_{ResNet}$ and $\theta_{GraphTrans}$ are from the encode module, $\theta_{GRU}$ and $\theta_h$ are from the GRU-based continual meta-Learning module, and $\theta_E$ is from the Bayesian graph edge labeling module.

---

**Algorithm 1** Continual Bayesian EGNN with Graph Transformer

---

**Input:**

A sequence of few-shot scene classification tasks $\{T\}$ with distribution $T \sim p(T)$, separated into sections of task periods $\{\{S_u, Q_u\}|u = 1, \cdots, U\}$.

**Ouput:**

Learnt parameters: ResNet Encoder $\theta_{ResNet}$, Graph Transformer $\theta_{GraphTrans}$. GRU module $\theta_{GRU}$ and hidden parameter $\theta_h$. $\theta_E$ of the Bayesian graph labeling.

**Details:**

1: **Initialize**: Load the ImageNet pretrained ResNet-18 weights. Initialize the parameters $\theta_{ResNet}, \theta_{GraphTrans}, \theta_{GRU}, \theta_h, \theta_E$.

2: **for all** $T \sim p(T)$ **do**

3:    **for all** $k \in \Omega$ **do**

4:       *Encoding Step:*

5:          Get features of input task samples as $X_T = X_S \cup X_Q$ by using ResNet.

6:       *Continual Learning Step:*

7:          **for** cell $c_i$ in task splits $\{c_1, ..., c_l\}$ **do**

8:             Calculate the structural attention $\mathbf{A}_{attn}$ and attention enhanced feature $\mathbf{x}'_i$ from ResNet feature $X_T$.

9:             *Iterative Continual Learning Step:*

10:             Calculate the historically enhanced feature $\tilde{\mathbf{x}}'_i$ from $\mathbf{x}'_i \in c_i$ by the GRU module.

11:             *Graph Edge Labeling Step:*

12:             Calculate plain feature difference based raw adjacency $\mathbf{A}_{sim}$ from enhanced features $\tilde{\mathbf{x}}'_i$, and get the Bayesian distributed adjacency parameters $\mu_{ij}^{\mathbf{W}}, \delta_{ij}^{\mathbf{W}}$, $\mu_{ij}^{\mathbf{b}}$ and $\mu_{ij}^{\mathbf{b}}$ by using CNN.

13:             Calculate the Bayesian distributed parameters $\mu_{ij}^{sim}$ and $\delta_{ij}^{sim}$, and generate the adjacency $\mathbf{A}_{final}^{sim}$, and get the final node correlations.

14:          **end for**

15:       *Update Step:*

16:          Calculate the edge loss $L_E$ and node loss $L_B$.

17:          Update the model parameters $\theta_{ResNet}, \theta_{GraphTrans}, \theta_{GRU}, \theta_h, \theta_E$ by $L_E + \gamma L_B$.

18:    **end for**

19: **end for**

---

## 6. Experiments and Results

In this chapter, the first section will briefly introduce datasets UC Merced Landuse [78], NWPU-RESISC45 [79], and AID [80], which are used for training and evaluation, which are commonly used datasets with increasing complexities that ensure the objective analysis of the behavior of the model. Section 6.1 goes through the details of experimental parameters, including the initialization parameters and settings of some of the crucial modules. Section 6.2 provides performance comparisons between our proposed model and the SOTA methods, discussing the pros and cons of the current model design. Section 7 then makes the ablation studies on the main performance-critical parameters of the model containing the classification accuracy, training sample ratio, etc., thus forming the detailed quantitative analysis on the main functional components.

### 6.1. Experiment Datasets and Experiment Setup

6.1.1. Datasets Description

To fully testify the crucial algorithmic properties including robustness and generality, the proposed algorithm has been fully verified on three challenging open access datasets: UC Merced Landuse, NWPU-RESISC45, and AID. The characteristics of these datasets and their training splits are introduced.

**UC Merced Landuse (UCM)** [78] is an open access scene classification dataset published on 28 October 2010 by the Merced laboratory, University of California. The dataset

contains 21 land use types, each containing at least 100 corresponding satellite images, all chosen from the USGS National Map Urban Area Imagery dataset, including a collection of urban and rural land types. In the dataset, each remote sensing image sized $256 \times 256$ with a ground sample distance of 0.3 m, with parts of the dataset image samples represented below.

In this article all datasets are split into three parts: Train, validation, and test, where the additional validation samples are for cross validation during the training iterations.

The splitting ratio of the partition on UC Merced dataset is 11:5:5, which is shown in Table 1 and Figure 6, with a categorical ratio of training at 52.38%. All three partitions contain natural and urban scenarios, which have imagery texture complexities ranging from simple to complex.

**Table 1.** Class segmentation for the UCM dataset.

| Train | Validate | Test |
|---|---|---|
| agricultural, beach, denseresidential, freeway, golfcourse, intersection, mediumresidential, parkinglot, river, runway, sparseresidential | baseballdiamond, buildings, forest, overpass, tenniscourt | airplane, chaparral, harbor, mobilehomepark, storagetanks |

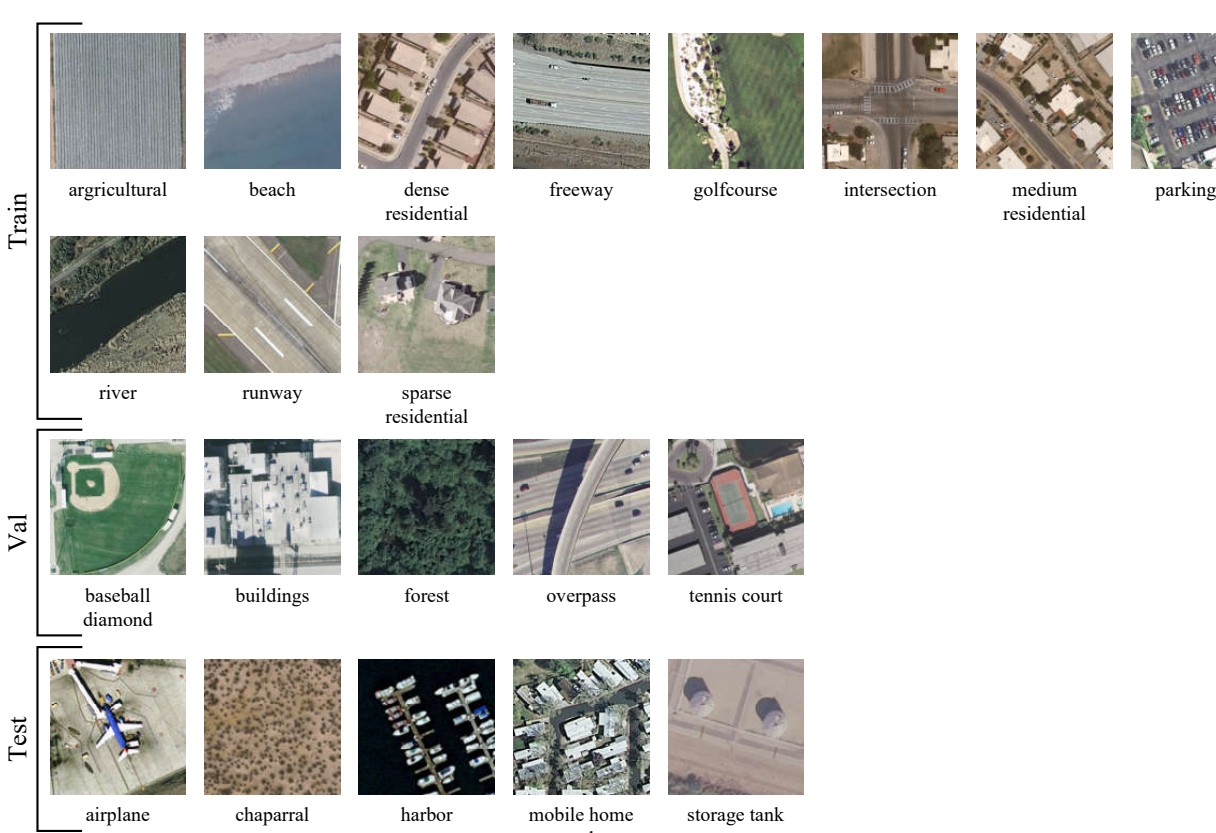

**Figure 6.** The train, validate, and test partitions of dataset UC Merced.

**NWPU-RESISC45** [79] is a public scene classification dataset published in 2017 by Northwestern Polytechnical University (NWPU). There are 45 scene categories containing an airport, tennis field, residual places, snow mountains, and seas, with 700 images in each

category collected from over 100 countries and regions all over the world, culminating to 41,500 images in total. Each image is in size $256 \times 256$ with a ground sample distance range from 0.2 m to 30 m.

The data sample split configuration of the NWPU-RESISC45 is shown in Table 2, the splitting ratios of the train, test, and validation partition are 25:10:10, which is shown in Table 2 and Figure 7, where 55.56% classes are chosen for training. Considering the complexity and data variances in NWPU-RESISC45, the scene categories selected for the train, test, and validation partitions are made as evenly as possible, by containing the typical natural and urban areas with similar image texture complexity distribution.

**Table 2.** Class segmentation for the NWPU-RESISC45 dataset.

| Train | Validate | Test |
|---|---|---|
| airplane, basketball_court, bridge, church, circular_farmland, dense_residential, forest, freeway, ground_track_field, industrial_area, intersection, island, meadow, medium_residential, mountain, overpass, palace, railway_station, rectangular_farmland, roundabout, runway, sea_ice, sparse_residential, tennis_court, terrace, wetland | baseball_diamond, chaparral, cloud, desert, mobile_home_park, palace, railway, ship, stadium, thermal_power_station | airport, beach, commercial_area, golf_course, harbor, lake, parking_lot, river, snowberg, storage_tank |

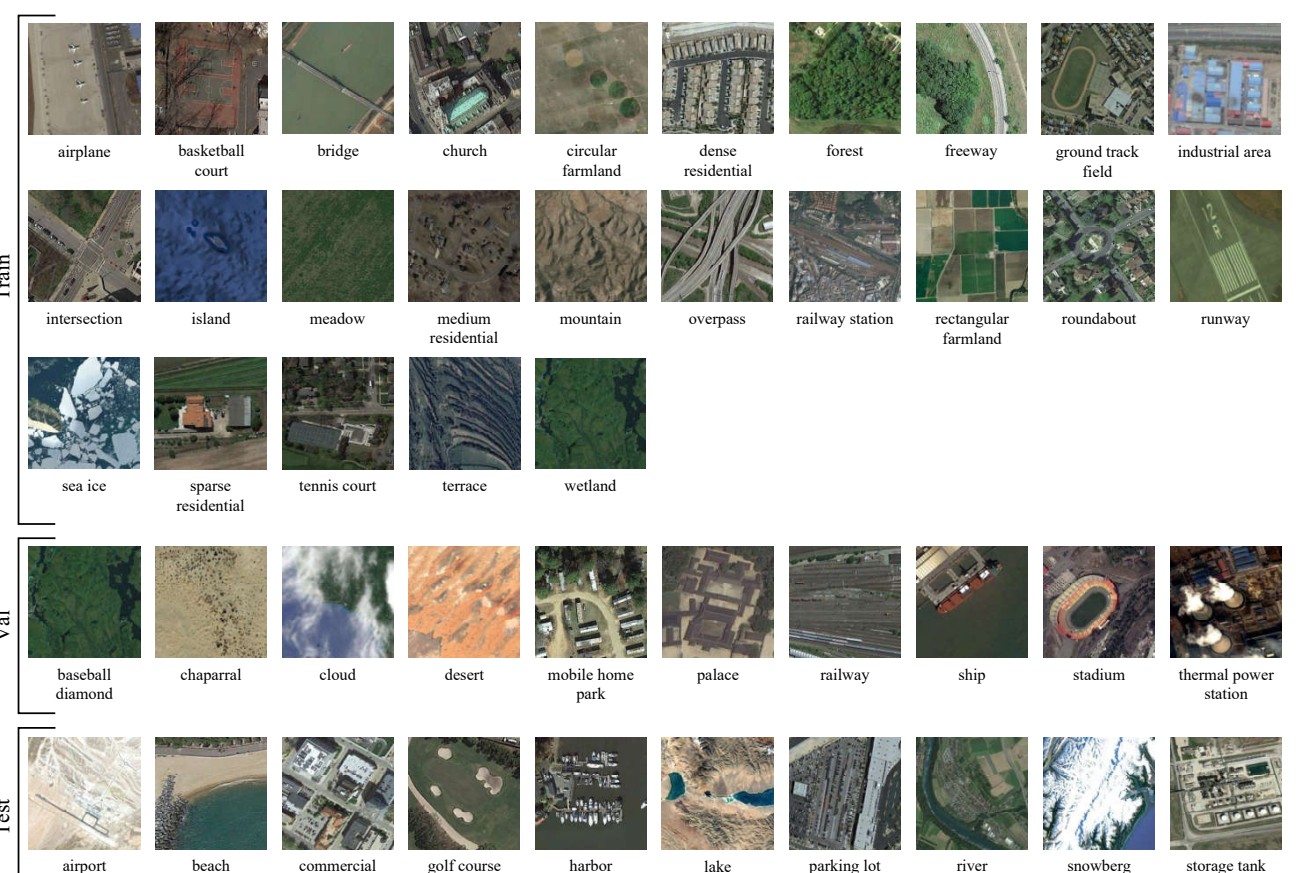

**Figure 7.** The train, validation, and test partitions of dataset NWPU-RESISC45.

The **Aerial Image Dataset (AID)** [80] is a public remote sensing scene classification dataset published in 2017 by Wuhan University and the Huazhong University of Science and Technology. It contains 30 scene classes of typical natural and man-made scenarios including an airport, sport stadium, river, meadow, and desert. Every scenery class has 200 to 400 images and adds up to a total quantity of 10,000. All images sized $600 \times 600$ and are selected by professional remote sensing researchers from Google Earth remote sensing collections with a ground sampling distance range from 0.3 m to 40 m.

The train, test, and validate splits of the AID scene classes is shown in the table with a ratio categorical quantity of 15:8:7, which is shown in Table 3 and Figure 8. Thus, the ratio of the training classes is 50%. And the scene class types in test and validation dataset distribute evenly in natural and urban regions, and the image texture complexity distributions are also similar.

**Table 3.** Class segmentation for the AID dataset.

| Train | Validate | Test |
|---|---|---|
| Airport, BaseballField, Center, Commercial, DenseResidential, Farmland, Meadow, Park, Parking, Pond, RailwayStation, School, SparseResidential, Stadium, StorageTanks | Bareland, Bridge, Church, Desert, Industrial, Mountain, Port, Square | Beach, Forest, Medium Residual, Playground, Resort, River, Viaduct |

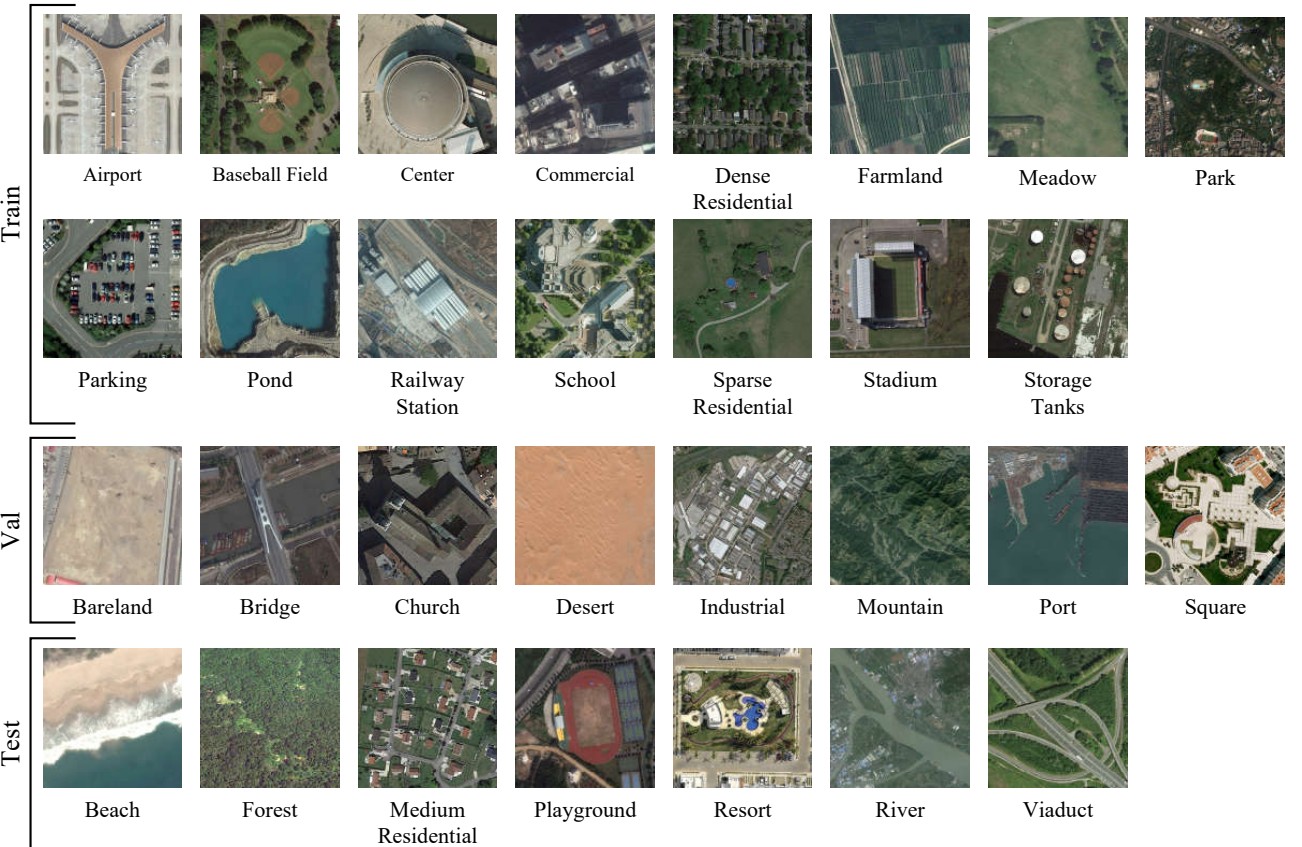

**Figure 8.** The train, validation, and test partitions of dataset AID.

### 6.1.2. Experiment Settings

For the **encode module** parameter settings, all input images fed to the ResNet-18 are scaled to a uniformed size of $256 \times 256$, and the size of the training image batch is 32 for comparisons in Section 6.2 for best performance, and 16 in Section 7. The encoded features from ResNet-18 is then mapped into 128 dimensional feature vectors which is then fed to the following graph transformer for attention oriented encoding. As to increase the diversity of the outputted module, the number of attention heads is set to 16 in the experiment, the dimension of the attention feature inside the graph transformer is set to 1024, as to ensure richer structural attention information will be introduced in to the new embedding space. In addition, the feature dimension of the attention-enhanced node feature outputted alongside the structural attention adjacency matrix is set to be 128, to be consistent with the input feature dimension.

For the **continual meta-learning module** parameter settings, in order to balance the computational cost and quality of encoded features, the iteration length of the GRU-based continual meta-learning is set to 16. Thus, there will be features from a single task with a batch size of 16, and features from 2 tasks for a batch size set to 32. Moreover, as the continual learning unit is an explicit distribution aggregation, the input and output feature dimensions are consistent with its direct predecessor and successor, which is 128 in the experimental settings.

**Bayesian graph labeling module** parameter settings: In the graph edge labeling module, the dimension of encoded edge-based correlation is set to 128, which is consistent with the dimension of features from the encoding module and the continual meta-learning module. Finally, the number of layers of the Bayesian edge labeling graph is set to 2, as the studies in ablation Section 7.3 suggested the best performance on all datasets.

**Training** and **optimization** settings: As the proposed algorithm contains multiple functional modules with varied feature encoding and correlating procedures, the Adam optimization method is adopted to cater with the high non-linearity characteristics. During training, the learning rate is initialized to $1 \times 10^{-3}$, with $1 \times 10^{-6}$ as the weight decay for every 15,000 iterations, and the gradient clip is set to 5.

### 6.1.3. Evaluation Metrics

Following the experimental settings, we compare the proposed algorithm with state-of-the-art counter-parts on datasets UC Merced, NWPU-RESISC45, and AID. The selected nine counterparts algorithms covers almost all the few shot scene classification algorithms verified on the three datasets, and are highly representative in their periods. For convenience, names of algorithms are abbreviated as: RS-MetaNet Ref. [53], Few-Shot Multi-Atten Ref. [31], Few-Shot Aerial Ref. [54], RS-SSKD Ref. [30], Know. Distill. Ref. [29], Proto. Calib. Ref. [27], SAFFNet Ref. [32], AMN Ref. [33], and ParamTrans Ref. [56].

In addition, in order to objectively illustrate the functional effects of the newly-introduced graph transformer structural attention encoding module and the modified Bayesian graph edge labeling module in the proposed algorithm, the continual meta-learning algorithm without these major changes is also included in comparison as the baseline algorithm, and denoted as Orig. CML-BGNN Ref. [52].

The classification accuracies and the usage ratios of training data are all listed in the Tables 4–6, the best and second-best accuracies are emphasized in bold and underlined. The accuracy metric being adopted in experiment analysis follows the definition of averaged accuracy described in SAFFNet Ref. [32] in detail, and is consistent with all the other counterpart algorithms.

### 6.2. Main Results

Based on the comparisons with counterparts and baseline, as the continual meta learning framework makes a better utilization of the historical prior, the averaged classification accuracies of our proposed and baseline algorithms stay almost always among the top 2 positions when compared to counterparts. Our proposed algorithm showed a minimum

9% improvement in accuracy, and a minimum 5% accuracy improvement over its continual meta-learning baseline on NWPU-RESISC45 and AID datasets.

**Table 4.** Classification accuracies on UC Merced.

| Method | Training Ratio | Backbone | Accuracy |
|---|---|---|---|
| RS-MetaNet [53] | 80.00% | ResNet50 | 53.57% |
| Few-Shot Multi-Atten. [31] | 76.19% | ResNet18 | 61.16% |
| SAFFNet [32] | 52.38% | ResNet18 | 65.89% |
| ParamTrans. [56] | 40.00% | ResNet12 | 62.96% |
| Orig. CML-BGNN [52] | 52.38% | ResNet18 | **89.13%** |
| Proposed | 52.38% | ResNet18 | <u>88.56%</u> |

Best and 2nd best accuracies are emphasize with **bold** font style and <u>underline</u>.

The complexity of the UC Merced dataset is slightly lower among the three datasets, as most algorithms produced a higher accuracy score on it. From Table 4, our proposed algorithm and its baseline occupied the first and second best positions on the list, with a leap of about 20% over algorithms using a standard meta-learning training mechanism. Besides, the accuracy of the baseline is even higher than our proposed version. This incidence is mainly caused by the dataset splits under our experiment settings, for there are only 5 classes in the test subset. By this, all tasks have the same collection of classes under the five-way one-shot meta-test setting, making the edge labeling graph structure almost static so that the flexibility structural pattern modeling effect of the graph transformer was weakened. Put it another way, such a phenomenon is also a convincing fact in that a structural attention calculation procedure is more beneficial in scenarios where a large number of novel classes and more complicated class relationships exists.

**Table 5.** Classification accuracies on NWPU-RESISC45.

| Method | Training Ratio | Backbone | Accuracy |
|---|---|---|---|
| RS-MetaNet [53] | 80.00% | ResNet50 | 46.32% |
| Few-Shot Aerial [31] | 55.56% | ResNet12 | 69.68% |
| RS-SSKD [30] | 55.56% | ResNet12 | 70.86% |
| Know. Distill. [29] | 62.22% | Conv-4 | 73.86% |
| Proto. Calib. [27] | 55.56% | - | 72.80% |
| SAFFNet [32] | 51.11% | ResNet18 | 64.63% |
| AMN [33] | 73.33% | ResNet18 | 74.25% |
| ParamTrans. [56] | 40.00% | ResNet12 | 67.14% |
| Orig. CML-BGNN [52] | 55.56% | ResNet18 | <u>85.63%</u> |
| Proposed | 55.56% | ResNet18 | **90.71%** |

Best and 2nd best accuracies are emphasize with **bold** font style and <u>underline</u>.

NWPU-RESISC45 is the biggest and most complex dataset among the three, with the highest number of classes and number of samples in each class, thus requiring higher inter and intra class discrimination modeling capabilities.

On it, due to the condensed categorical sample feature distributions by the feature rectification and smoothing effects through the continual meta-learning procedure, our proposed algorithm and its baseline achieved the top-1 and top-2 accuracies with a leap of more than 10% over the counterpart algorithms. Besides, the accuracies of almost every algorithm increased compared with scores on UC Merced, which mainly benefit from the richer supply of training samples in each class. Moreover, our proposed algorithm overpassed the baseline Orig. CML-BGNN Ref. [52], as the graphic structural attention calculation mechanism began to show its advantages in modeling more flexible graph structure caused by the increased quantity of categorical samples and number of classes,

which can promote intrinsic intermediate class correlations through the attention enhanced edges in longer node connection hops.

**Table 6.** Classification accuracies on AID.

| Method | Training Ratio | Backbone | Accuracy |
|---|---|---|---|
| RS-MetaNet [53] | 80.00% | ResNet50 | 54.26% |
| Few-Shot Multi-Atten. [31] | 80.00% | ResNet18 | 74.52% |
| Know. Distill. [29] | 43.33% | Conv-4 | <u>78.47%</u> |
| SAFFNet [32] | 50.00% | ResNet18 | 67.88% |
| ParamTrans. [56] | 40.00% | ResNet12 | 77.15% |
| Orig. CML-BGNN [52] | 50.00% | ResNet18 | 71.35% |
| Proposed | 50.00% | ResNet18 | **87.60%** |

Best and 2nd best accuracies are emphasize with **bold** font style and <u>underline</u>.

Among the three datasets, AID has the median number of classes and median categorical sample size. Such characteristics reduces the complexity of class correlations and is beneficial for methods with strong categorical feature encoding capability, and the data distribution stability in each category is increased. As a result of this, the accuracies of algorithms Few-Shot Multi-Atten. Ref. [31], ParamTrans. Ref. [56], and Know. Distill. Ref. [29] even surpassed our baseline algorithm Orig. CML-BGNN Ref. [52] using a continual meta-learning mechanism. Specifically, compared with Orig. CML-BGNN Ref. [52], Few-Shot Multi-Atten. Ref. [31] is advantageous in its adoption of an attention calculation mechanism, which is more suitable for occasions where greater inner class diversities exist. For ParamTrans. Ref. [56] and Know. Distill. Ref. [29], their superiorities came from the richness of samples for their offline knowledge transfer mechanism, which helps greatly in strengthening the stability and generalization capability from larger base training datasets and sophisticated teacher models. For our proposed algorithm, its main characteristics is the combinatorial usage of structural attention encoding via the graph transformer, and the online feature smoothing via a continual meta-learning mechanism. These characteristics facilitate the smoothing of the fluctuations in distributions of categorical features in embedded space, which boosts the performance of the proposed algorithm to surpass most of the state-of-art counterparts.

*6.3. Knowledge Transition Efficiency Analysis*

For meta-learning algorithms, data-to-knowledge transition efficiency is an important performance index, and we approximate it here via an empirical measurement as a ratio between the averaged accuracy of the model and the dataset usage. The dataset usage is defined as the ratio between the number of classes used in meta-training and the overall dataset class quantity, which is conceptually consistent with most counterpart algorithms.

Based on the definition, the knowledge transition efficiency of algorithms can be visually shown on a scatter plot, with the X-axis marking data usage and the Y-axis marking the algorithm evaluation accuracy. In this plot, algorithms with a higher accuracy and lower data usage would stay close to the upper left side.

In Figure 9, the efficiency evaluation result on a different dataset is marked with a different marker style. As shown, our proposed algorithm and its continual meta-learning baseline stayed at positions (within the dashed line circle) within the upper left region of the plot, indicating a good trade off between the training data quantity and overall performance on all three datasets. Such high efficiency comes from the effective online knowledge transfer through the continual meta-learning from sequences of meta-test tasks. Besides, Know. Distill Ref. [29] and ParamsTrans Ref. [56]. also achieved a highly knowledge transition ratio among the counterparts. These two algorithms showed its superiority of offline knowledge transfer as they surpassed the baseline Orig. CML-BGNN Ref. [52] on dataset AID, which proved to be efficient enough within a moderate class relationship and sample distribution complexity range. Through such a comparison, a continual meta-

learning mechanism as well as its combination with graph structural attention calculation, proved to be a promising novel meta-learning technique for extending the flexibility of the model on application scenarios with a greater number of classes and higher quantities of sample distributions.

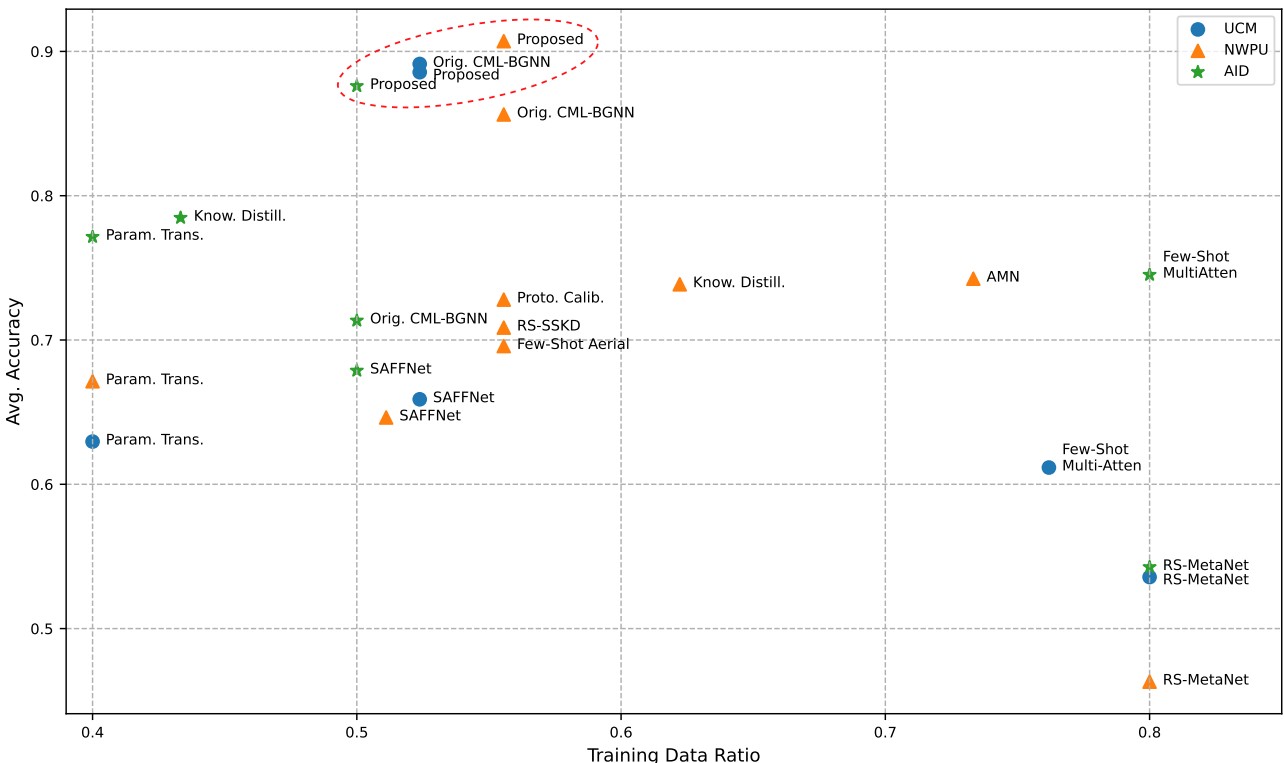

**Figure 9.** Data-to-knowledge transition efficiency comparison scatter plot. The positions of proposed algorithms are marked with red dashed circle.

### 6.4. Classification Accuracy Details

Detailed classification result of our proposed algorithm is illustrated by the confusion matrices, as in Figure 10, showing the confusion matrices on datasets UC Merced, NWPU-RESISC45, and AID.

Interestingly, accuracies of the classes are rather close, which have a very small difference between each other. One major reason is the usage of an edge labeling graph for class discrimination and classification. In an edge labeling graph, classification depends on the correlation modeling through the weight of the edges. So suppose there is a class A with a high classification accuracy, then under the N-way K-shot meta-learning settings, that means some of the remaining $N-1$ categorical correlations given by the edges are considerably correct. Moreover, node correlations in graph models can be inferred through an intermediate connection at more distant edge connections, so such an accuracy can be propagated onto classes closer in probability distributions.

In Figure 10a, on dataset UC Merced, Chaparral is the second best class as it is the only natural scenario compared with others containing man-made objects. The land type storagetanks has the lowest accuracy as it can be found near a harbor, airport, and dense urban regions, thus a classifier made a high false positive rate on other land types except chaparral.

In Figure 10b, on dataset NWPU-RESISC45, accuracy differences between classes are slightly higher than on UC Merced, as there are more classes involved in the test, and some of them are not within the 2-hop length correlation with the best classified one. The best classified class golf_course reached an accuracy of 91.82%, as the composition of golf_course is generally simple, mainly consisting of sand pot and large meadows. While

the storage_tank still remained at a lower level with large false positive confusion ratios with other land types.

In Figure 10c, on dataset AID, the best classified land type is resort with an accuracy of 88.35%, whose contents are plants, water bodies, and buildings mainly exists in park-like regions, whose texture patterns and shapes are very different from those in natural regions or normal residential areas, thus the false positive ratio on this land type is at a rather low rate.

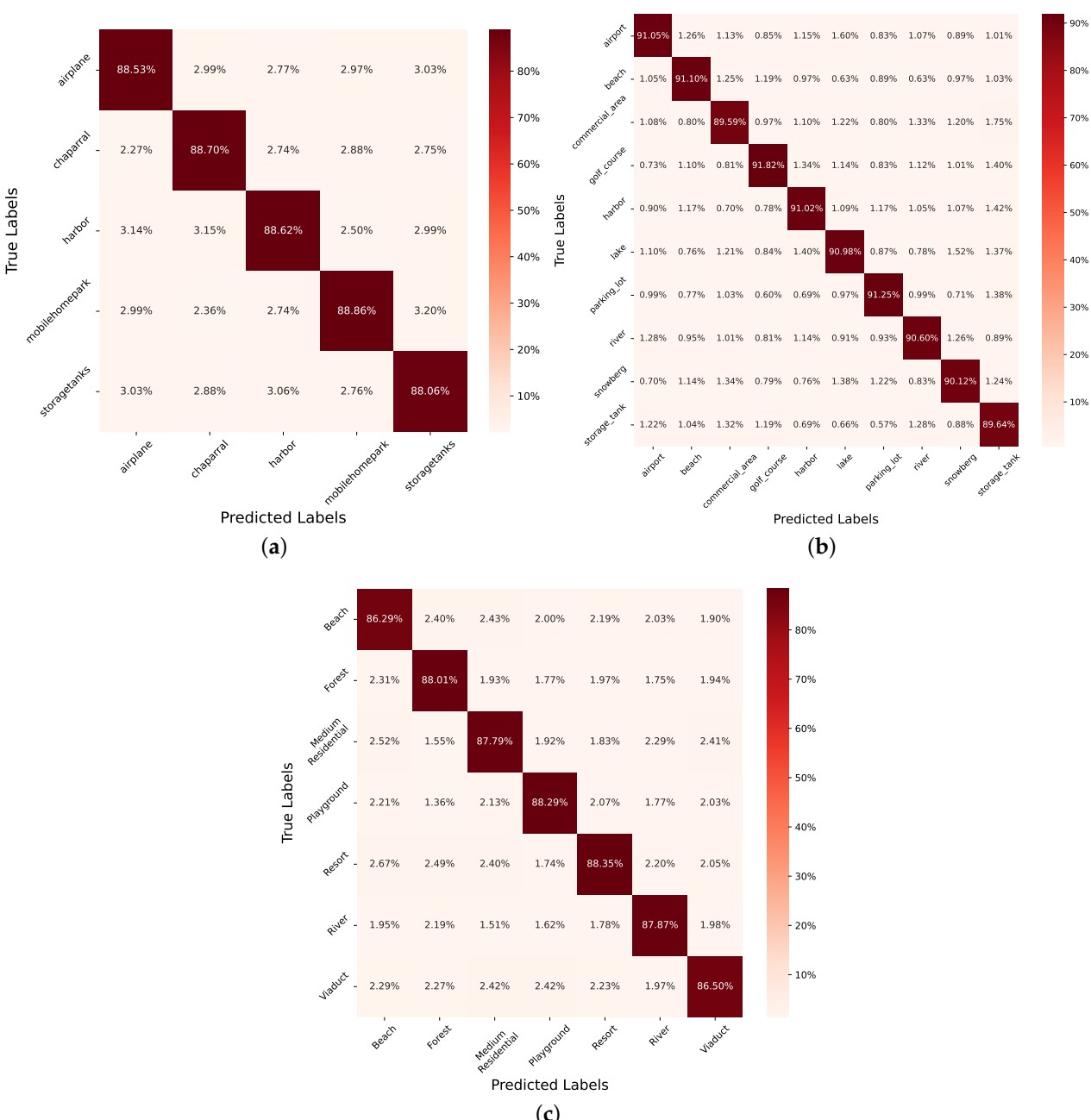

**Figure 10.** Classification confusion matrix on datasets. (**a**) Classification confusion matrix on dataset UC Merced; (**b**) Classification confusion matrix on dataset NWPU-RESISC45; (**c**) Classification confusion matrix on dataset AID.

## 6.5. Training Stability Analysis

In this subsection, we analyze the generalization and stability pattern of our proposed model on the three datasets, by comparing the changes in evaluation accuracies and

evaluation losses during training to that of the continual meta-learning baseline model Orig. CML-BGNN Ref. [52]. Generally speaking, a robust model tends to have a more stable evolution pattern during training, where the changes in evaluation losses and evaluation are gradual and slow, which have very small fluctuations through the training process.

Figure 11 shows a comparison of changes in accuracies and losses between our proposed model and its baseline Orig. CML-BGNN Ref. [52]. From the figure, it can obviously be seen that our proposed model has smaller evaluation accuracy oscillations during the training process, and are mostly higher than the scores produced by the baseline algorithm except at the UC Merced dataset. In training losses, changes in the losses of our proposed algorithm are also smaller than those of the baseline algorithm, and always stay at a lower level except in the UC Merced dataset. The main reasons for the superiority of the baseline algorithm has been explained in Section 6.2, which is mainly caused by the fitness between data split and meta-test settings. So, according to the performance comparisons and the overall accuracies constrasting in Tables 4–6, the proposed algorithm had made better classification performances on NWPU-RESISC45 and AID, and shows a comparative accuracy on the UC Merced dataset.

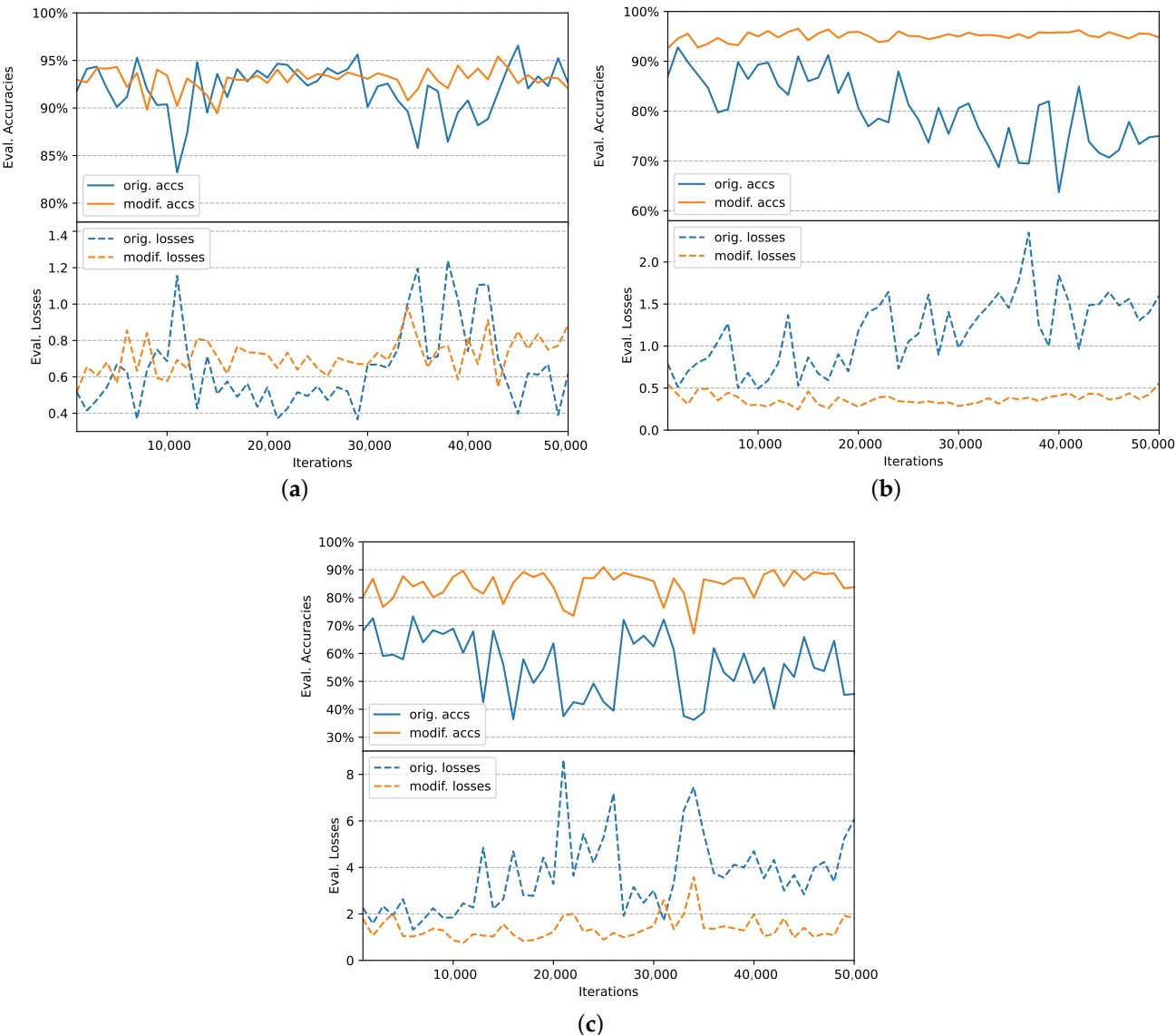

**Figure 11.** Evaluation accuracies and losses comparisons with the baseline model during training. (**a**) UC Merced dataset; (**b**) NWPU-RESISC45 dataset; (**c**) AID dataset.

## 7. Ablation Studies

### 7.1. Graph Transformer Heads

For the graph transformer-based graphic structure attention encoding module, the number of attention heads determine the quantity of significant structures it is capable to encode; and the higher the number of encoded structural pattern quantity, the greater the complexities in the graphic structure and categorical correlations it is able to describe.

Table 7 compares the classification performances of our proposed model on three datasets with different attention head quantities. From the table, an obvious equilibrium between the head quantity and dataset complexity can be seen. For UC Merced and NWPU-RESISC45 with fewer samples in each class, the classification accuracies declined as the number of heads increases, but at a rather small rate within 3%. While on AID, which has much more samples in each class, the increase of head quantity greatly boosts the performance with a rate of over 6%. So increasing the head numbers is an effective model tuning method when the inner class sample variances are large.

**Table 7.** Influences of the number of attention heads of the graph transformer on the performance.

| Num. Heads | Classification Accuracy | | |
| --- | --- | --- | --- |
| | UC Merced | NWPU-RESISC45 | AID |
| Num. Heads 4 | 86.59% | 87.53% | 71.67% |
| Num. Heads 8 | 84.88% | 85.08% | 77.77% |
| Num. Heads 16 | 81.79% | 83.34% | 85.03% |

In Figures 12 and 13, we show the classification confusion matrices and their difference with graph attention heads number equaling 4 and 16. In each figure, the left two sub-figures are confusion matrices with different head numbers, and the third one on the right is the difference matrix between the second and first confusion matrices. In the confusion matrices, differences matrix of the third sub-figure, categorical accuracies, and the false positive rates can be easily seen.

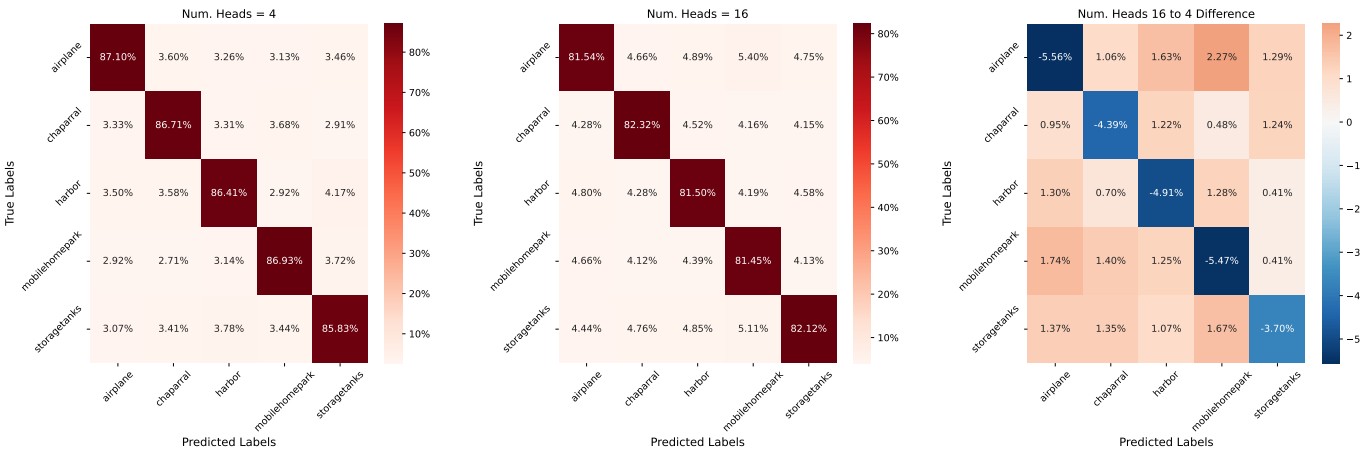

**Figure 12.** Confusion matrices and their difference on dataset UC Merced with graph transformer attention heads equaling 4 and 16.

In Figure 12, on dataset UC Merced, as there are only five test classes and the edge labeling graph structure is fixed, using more attention heads will make the model more complex, thus becoming more prone to overfitting during training. Therefore, the false positive rates between complicated man-made scenarios such as mobile-home-park to airplane, storage-tanks to mobile-home-parks increases greatly, while the accuracy loss on a natural scenario chaparral is smaller.

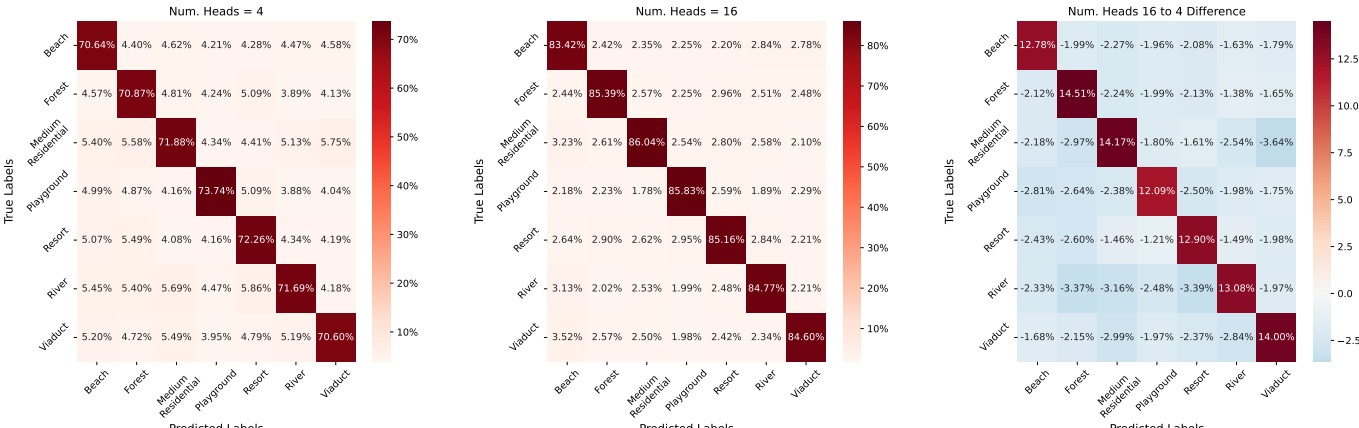

**Figure 13.** Confusion matrices and their difference on dataset AID with graph transformer attention heads equaling 4 and 16.

In Figure 13, on dataset AID, an increased dataset complexity is fit for the graphic structural attention encoding mechanism where a greater modeling flexibility is required, thus the discriminative power increased in distinguishing complex scenery differences including river vs forest, river vs medium residential, viaduct vs medium residential etc. The added structural attention shown fitted for modeling the local texture patterns, making confusing classes easier for recognition.

### 7.2. Length of Continual Meta-Learning Iterations

The Continual Meta-learning Iterations Length (CMLL) is a strong control variable for the capability of online historical meta-testing knowledge utilization, where a longer iteration length ensures greater connectivity and more active information sharing between tasks previously processed in the continual meta-learning sequence.

Table 8 shows the classification accuracies on the three datasets with different continual meta learning iteration lengths. As the iteration goes longer, the changes in classification accuracies is roughly linear. Great leaps in accuracy improvements can be observed as the iteration length changed from 4 to 16 on all datasets, while a small accuracy decrease appeared as the length grows from 4 to 8. Such performance degradation is caused by the hidden embedding feature smoothing effect in the continual meta-learning mechanism, which happened when the distribution of categorical features is shrunk by averaging, while not strong enough to enlarge inter-class discrepancies but condensed the distribution density of samples in the confusing interlaced region between classes. Obviously, the iteration length of 16 is more appropriate for concentrating the categorical features.

**Table 8.** Influences of the number of iterations of the GRU-CML module on performance.

| Num. Iter.(s) | Classification Accuracy | | |
| --- | --- | --- | --- |
| | UC Merced | NWPU-RESISC45 | AID |
| Num. Iter.(s) 4 | 76.56% | 69.84% | 75.02% |
| Num. Iter.(s) 8 | 72.80% | 75.40% | 73.36% |
| Num. Iter.(s) 16 | 81.79% | 83.34% | 85.03% |

The detailed classification accuracy changes from length 4 to 8 and from length 4 to 16 are shown in Figures 14 and 15. As can be seen, a large false positive rate increments and decrements mainly happened on classes including complex man-made land types, for instance playground vs medium residential and viaduct vs forest, where greater feature space interlacing exists between them caused by the shared complex ground structures and similar ground objects. Class pairs with significant structural differences are lesser influenced by the changes in continual meta learning length, as the semantic confusion between them is weaker.

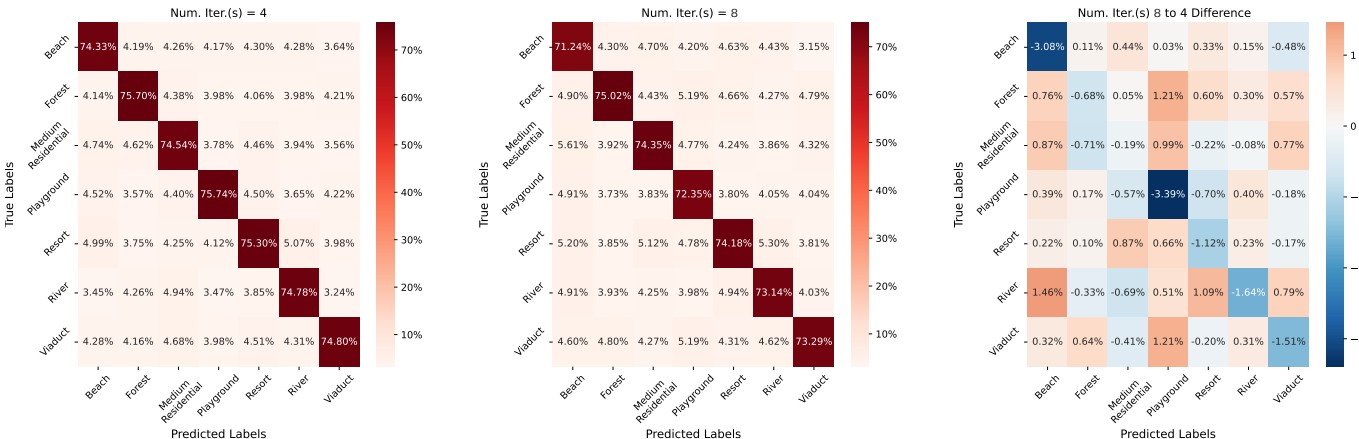

**Figure 14.** Confusion matrices and their difference on dataset AID with continual meta-learning iteration lengths 4 and 8.

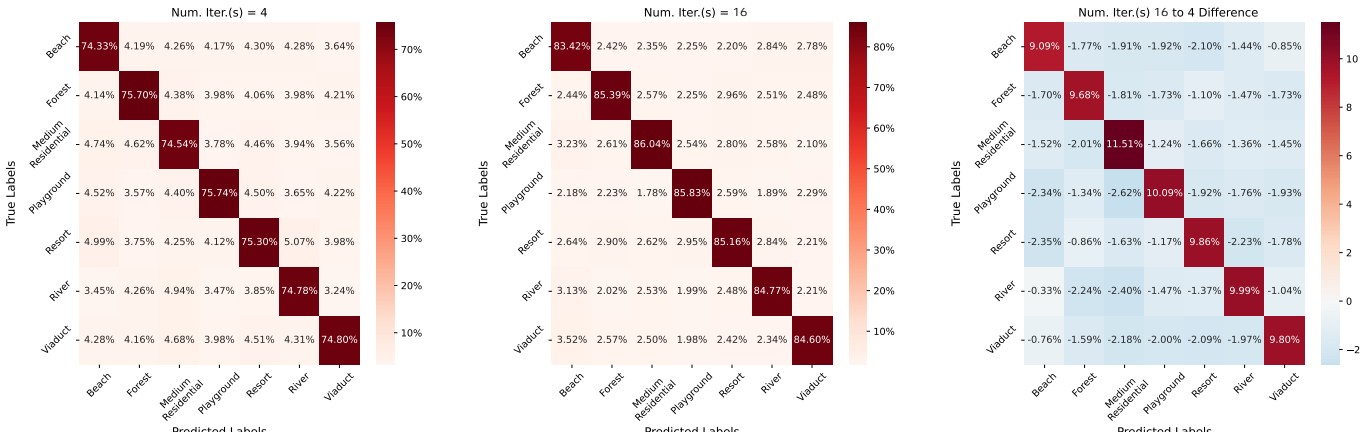

**Figure 15.** Confusion matrices and their difference on dataset AID with continual meta-learning iteration lengths 4 and 16.

Finally, the changes in class accuracies are illustrated as bar plots in Figure 16. It can be seen that classes having higher accuracies in length 4 setting tend to keep its rank in the 16 setting, having more increments in accuracy. Thus, enlarging the iteration length of continual learning module is more like an equal scaling of the classification discriminative power, which will not change the relation positions of categorical features in the embedded feature space.

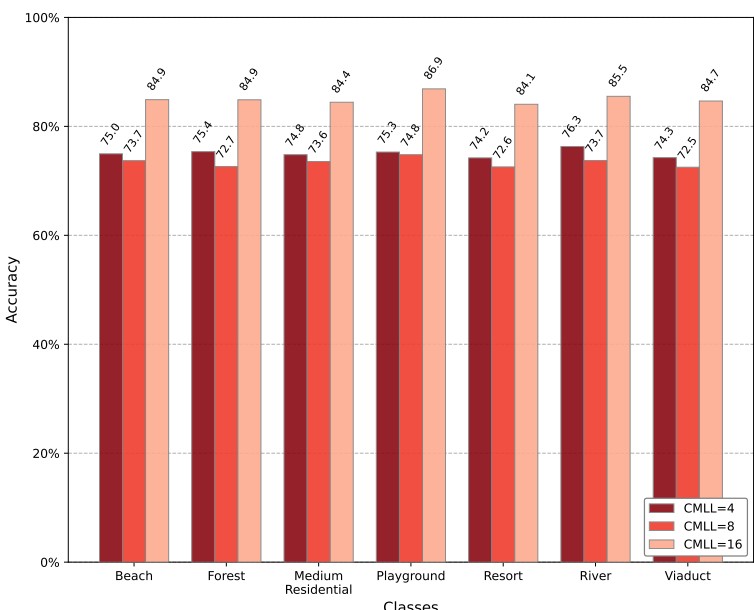

**Figure 16.** Comparison of class accuracies when different Continual Meta-learning Iteration Lengths (CMLL) are 4, 8, and 16.

### 7.3. Number of Layers in Bayesian Edge Labeling Graph

The number of layers in the Bayesian edge labeling graph module decides the modeling capabilities for complex node feature correlations, so there is also an equilibrium that needs to be balanced between the complexities of the dataset and model.

Table 9 shows the contrasts in classification accuracies on the three test datasets when different a Bayesian edge labeling graph layer number is chosen. It can be easily observed that the best performance of our model appears with the choice of 2 layers, where the data and model complexities matched well.

**Table 9.** Influences of the number of Bayesian edge labeling graph layers on performance.

| Num. Layers | Classification Accuracy | | |
| --- | --- | --- | --- |
| | UC Merced | NWPU-RESISC45 | AID |
| Num. Layers 1 | 76.76% | 82.91% | 83.73% |
| Num. Layers 2 | 88.56% | 90.71% | 87.60% |
| Num. Layers 3 | 81.79% | 83.34% | 85.03% |

Figure 17 shows the confusion matrices on dataset AID when the graph labeling model layer quantities are set to 1, 2, and 3. With the number of layers increased, the false positive ratios increased almost monotonic in classes such as playground and beach, where differences in the scenario content exists in multiple spatial scales. While for classes sharing partial scenario contents or ground object types, such as a playground and resort, using a deeper graph labeling layer depth only increases the semantic confusions between classes, which jeopardize the overall performance.

Finally, the varied accuracies of all classes using different layer quantities is shown in Figure 18 for explicit comparisons. In it, land types with significant content differences from others, such as beach and viaduct, their classification accuracies tend to be less affected by the changes in the graph labeling layers.

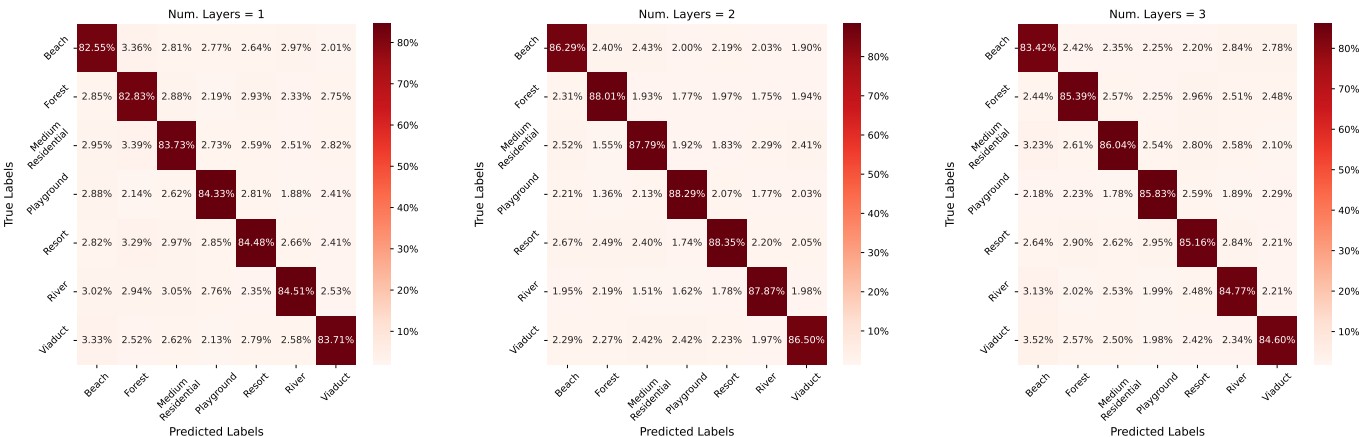

**Figure 17.** Confusion matrices comparison between settings using a different quantity of Bayesian edge labeling graph layers.

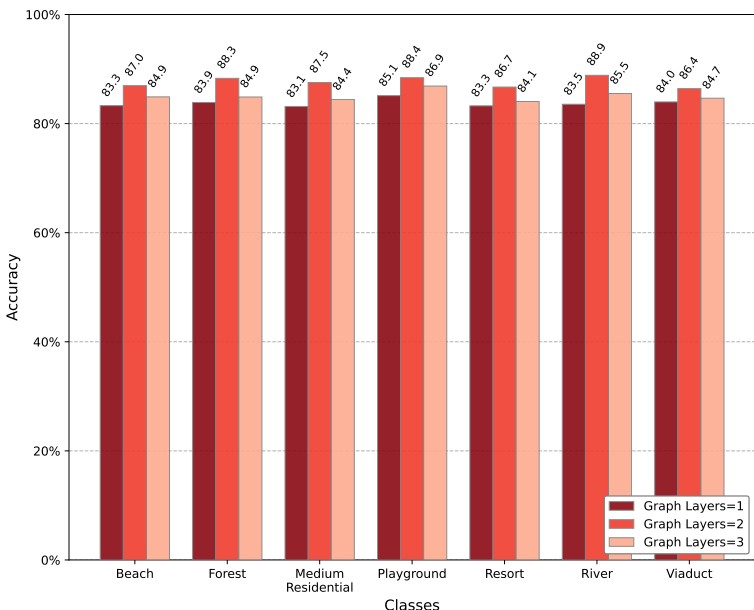

**Figure 18.** Comparison of class accuracies when a different number of Bayesian edge labeling graph layers is used.

## 8. Conclusions

As the demands of intelligent remote sensing imagery interpretation increases, the necessity of developing fast model adaption methods that is less dependent on human supervision and a smaller dataset grows significantly. Few-shot remote sensing scene classification is one such fundamental technique that is indispensable in a large variety of tasks. Among the popular few-shot learning methods, this article studies efficient algorithms based on the continual meta-learning principle, which is derived from meta-learning methods, but can alleviate the data isolation in standard meta-learning training schema by promoting the utilization of historical prior during the online meta-testing procedure. The proposed algorithm introduced a novel and efficient composition of latest techniques, including graph neural network-based classification and graph transformer-based structural attention encoding under the continual learning framework, which can significantly improve flexibility in modeling the complex categorical structure under data urgent conditions. In the proposed algorithm structure, node features encoded from support and query samples are further enhanced by the graph transformer calculating structural attention for better intra and inter class discrimination. The GRU-based continual meta-learning module aggregates the distribution of node features to the class centers and

enlarges the categorical discrepancies. In addition, the node correlations modeled by edge weights are rectified by both the primitive node feature distances and the Bayesian-style Gaussian estimations. The advantages of the proposed model are clearly illustrated through the comparisons with other standard meta-learning-based counterparts, where a minimum of 9% leap in accuracy can be observed. The effectiveness of the novel combination structural attention computation being proposed is shown by the comparison with the baseline algorithm. Despite this, much work still needs to be done in further studies. Our future work will focus on improving the flexibility and scalability of the continual meta-learning-based tasks period sampling schema, thus making the algorithm fit for a wider range of dataset sizes and complexities, and is more useful for real life applications.

**Author Contributions:** Conceptualization, F.L. and H.C.; methodology, F.L. and S.L.; software, F.L. and S.L.; validation, F.L. and X.L.; formal analysis, F.L.; investigation, F.L. and S.L.; resources, F.L.; data curation, S.L.; writing—original draft preparation, F.L., S.L. and X.F.; writing—review and editing, F.L.; visualization, F.L.; supervision, H.C.; project administration, F.L.; funding acquisition, F.L. All authors have read and agreed to the published version of the manuscript.

**Funding:** This research was funded by National Natural Science Foundation of China (No. 61806199).

**Data Availability Statement:** Not applicable.

**Acknowledgments:** We would like to thank Section Managing Editor Chryssa Liu, Assistant Editor Trista Yang, Assistant Editor Hope Zhu, Assistant Editor Adriana Wang, for their patience and responsibility in processing our manuscript and sending notification emails. We would also like to thank the professors for using their valuable time during the end of semester to review our manuscript and give so many constructive comments.

**Conflicts of Interest:** The authors declare no conflict of interests.

## Abbreviations

The following abbreviations are used in this manuscript:

| | |
|---|---|
| AID | Aerial Image Dataset |
| BGNN | Bayesian Graph Neural Network |
| CML | Continual Meta-Learning |
| CNN | Convolutional Neural Network |
| DLR | German Aerospace Center |
| FSC | Few-Shot Classification |
| FSL | Few-Shot Learning |
| GNN | Graph Neural Network |
| GPN | Gated Propagation Network |
| GRU | Gate Recurrent Unit |
| HGNN | Hierarchical Graph Neural Network |
| INRIA | The National Institute for Research in Computer Science and Automation |
| LSTM | Long Short-Term Memory |
| MAML | Model Agnostic Meta-Learning |
| ML | Meta-Learning |
| NWPU | Northwestern Polytechnical University |
| RESISC | Remote Sensing Image Scene Classification |
| RNN | Recurrent Neural Network |
| SAM | Self-Attention Meta-Learner |
| UCM | UC Merced Landuse Dataset |

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
