# Peer review of "Structural Attention Enhanced Continual Meta-Learning for Graph Edge Labeling Based Few-Shot Remote Sensing Scene Classification"

_remotesensing, doi:10.3390/rs14030485_

Round 1

Reviewer 1 Report

In this paper, the authors present a novel approach to few-shot scene classification based on a continual meta-learning algorithm, incorporating structural attention using Graph Transformers to increase the classifier's discriminative power. The paper compared the proposed approach against state-of-the-arth baseline methods on three remote sensing scene classification datasets and showed promising results. However, I have some few questions, clarifications, and suggestions below that I believe must be addressed to improve the quality and the readability of the paper.

In lines 369 to 372, the authors described the GRU and Bayesian Graph, feature optimization and node adjacency calculations, are performed in a "layer-wised" form. However this form is not really apparent in the figure in referred to. Can this be further expounded using notations or equations?

In lines 395 to 396, s r^{enhc) different from either r^(adj) and r^(final), from the figure and the notation it seems to be the same as the former?

In Algorithm 1, line 10, shouldn't c_{j} be c_{i}?

In Algorithm 1, line 16, do the edge and node losses correspond to L_{B} and L_{E} in Equation 15?

In subsection 6.1.2, lines 468 to 478, where there any hyperparameters of the encode module that were varied to see any significant effect on performance, like the ablations experiments done on the Graph Transformer and Bayesian Edge Labeling Graph hyperparameters? Or all/most were taken from settings of the architectures that were used in the reference papers?

Tables 4 onwards, for the sake of clarity, can you please give a short (one-sentence or so) description of what the training ratio is?

Lines 523 to 524, debatable, does high mean > 60% in the context of few-shot classification?

The sentence in lines 527 to 529, it is not clear whether the accuracy numbers shown in the table are using the same classes on the training, validation and testing sets, at least the ones with the same training ratio. Could you please clarify this? And if they are not using the same set of classes, explain why.

Line 534, what does "high class quantity" mean? larger N?

Line 536, what is the difference between "categorical" and "category" quantities? Can you clarify both terms? Do they mean the number of classes and number of samples in each class correspondingly?

Figure 9, couldn't this transition efficiency measure be reduced as just another column in the accuracy tables above? E.g. take the ratio of the values of the two axis. Seems like labeling of the models in the scatterplot forced the figure to be too big and hence a lot of whitespace.

In Table 8, why isn't the iteration length further increased? There still seems to have a massive gain with doubling it from 8 to 16.

Line 709, results for 2 layers is not shown in the table.

Line 712, again the values are not consistent with what is shown in the talbe, please explain.

Some rephrasing and/or correction of typographical and grammatical mistakes are needed for the following portions of the manuscript.
Line 32 "with fast emergency"
Line 45 "made to alleviated"
Line 68 "as most of they dropped"
Line 131 "as closing to global optimal parameters"
Sentence in lines 175 to 176
Line 178 "they expert at extracting"
Line 246 "a portion of meta-learning researching"
Line 248 "model updating throughout the processing"
Line 249 change the index "p" to a different notation
Line 256 "time serial sample modeling techniques", do you mean time series sampling techniques?
Lines 264 to 265 "enhance the optimizing the generalization ability"
Line 290 to 291 "It core component"
Line 293 to 294 "sequentially encode node features"
Line 318 "node feature differing"
Line 344 remove "gotten"
Algorithm 1, Line 12, "CNN networks" is redundant, I'd say CNN or convolutional networks is enough.
Line 418 "including the intializing", including the initialization
Line 419 "makes performance compare", provides a performance comparison
Line 529 "all tasks carray"
Line 555 "is advantages"
The sentence in lines 560 to 564, please rephrase the statement for clarity and grammatical correction
Line 596 "very small different between"
Line 610 "is slightly", are slightly
Line 663 "thus became more", thus becoming more
Line 672 "confusion classes", confusing classes
Line 713 "lasses such as"
Line 721 "affective by", affected by

Author Response

Dear professor.
Thank you for providing so many constructive suggestions to improve our work. The detailed replies are represented in the attachment.
We are very thankful for your patience in helping to find out a good number of typos and unsuitable phrases. We are very sorry for having brought you so many troubles in reading. Your major comments including expounding the descriptions on "layer-wised" adjacency calculation have been carefully replied, and we have also pointed out the rectified texts positions in the revised version of the manuscript.

Reviewer 2 Report

The authors propose a new method for few-shot scene classification based on a meta-learning principle. This approach provides an increase in classification accuracy by 9%.

Notes for work:

1) The article mentions methods of image augmentation using generative models, but does not mention simpler algorithms that also allow improving the quality of training (10.1109 / SYNCHROINFO49631.2020.9166000, https://doi.org/10.3390/info11020125)

2) Line 91: a sentence cannot start with [50] ... It is better to use “Article [50] ...”

3) Line 94: a sentence cannot start with [51] ...

4) Tables 4-6 have a Training Ratio column. It would be interesting to compare the results when changing this parameter for the proposed method.

5) Table 9 shows the number of layers 4, 8 and 16. But we are talking about 1,2 and 3 layers.

6) The section "Acknowledgments" should be deleted if there is no information there.

Author Response

Dear professor, thank you for your general approval of our work, the detailed reply and explainations are provided in the attachment. We have answered carefully to your comments, and revised our manuscript according to your suggestions. 

Reviewer 3 Report

The study focuses on improving the efficiency of scene classification based on the Few-Shot approach. The authors introduce a novel few-shot scene classification algorithm based on continual meta-learning as well as a new Graph Transformer. The new approach fuses more historical prior knowledge from a sequence of tasks within sections of meta-training/meta-testing periods and is expected to enhance the inter-task correlation and increase the discriminative power between classes. Overall, the manuscript is well written with clear objectives, methods, and analysis. However, the manuscript is lengthy, and I suggest the authors summarize some parts of the method and results section (highlighted in the annotated pdf). For example, section 2 can be summarized and section 5 can be put as an appendix.  Besides this, I have no major concern about the manuscript, and there are some minor issues that I highlighted in the annotated pdf.  

Author Response

Dear professor, thank you for your careful reading of our manuscript. We have carefully answered your questions in the PDF attachment, and have revised our manuscript carefully according to your suggestions. Specifically, we have summerized section 2, and suppressed section 5.1 and cut off the well-known informations for succinctness.

The revised manuscript cannot be uploaded via this link, and we have sent it to editor. I think the revision might be distributed to you later.

Reviewer 4 Report

This paper presents a new remote sensing (RS) scene classification method under the few-shot scenario, in which a graph transformer attention block, a continual learning block, and a Bayesian graph correlation modeling block are combined. Overall, this work is interesting and feasible, and sufficient experiments illustrate the effectiveness of the proposed method. Only some minor issues are summarized as follows.

  1. How about the time costs of your method?
  2. Why do you fix your experiments' training, validating, and testing classes? Is that fair?
  3. How do you conduct the compared models? Do you use the same experimental settings?
  4. The influence of Eq. 15 should be studied for readers.

Author Response

Dear professor, thank you for your patient reading of our manuscript. The replies have been written in the PDF attachment. Your comments and suggestions are very constructive, and we have answered carefully. The manuscript has also been carefully revised according to your suggestions.

The PDF attachment only contains the answers. Our revised manuscript has been sent to editor, so we think it will be distributed very soon.
